# Prehistoric population expansion in Central Asia promoted by the Altai Holocene Climatic Optimum

Lixiong Xiang [1], Xiaozhong Huang [1] ✉, Mingjie Sun[1,2], Virginia N. Panizzo [2] ✉, Chong Huang[1], Min Zheng [1], Xuemei Chen[3] & Fahu Chen[4,5]

How climate change in the middle to late Holocene has influenced the early human migrations in Central Asian Steppe remains poorly understood. To address this issue, we reconstructed a multiproxy-based Holocene climate history from the sediments of Kanas Lake and neighboring Tiewaike Lake in the southern Altai Mountains. The results show an exceptionally warm climate during ~6.5–3.6 kyr is indicated by the silicon isotope composition of diatom silica ($\delta^{30}Si_{diatom}$) and the biogenic silica (BSi) content. During 4.7-4.3 kyr, a peak in $\delta^{30}Si_{diatom}$ reflects enhanced lake thermal stratification and periodic nutrient limitation as indicated by concomitant decreasing BSi content. Our geochemical results indicate a significantly warm and wet climate in the Altai Mountain region during 6.5–3.6 kyr, corresponding to the Altai Holocene Climatic Optimum (AHCO), which is critical for promoting prehistoric human population expansion and intensified cultural exchanges across the Central Asian steppe during the Bronze Age.

The Altai-Sayan region is the geographical heartland of early cultural contacts between eastern and western prehistoric Eurasia, which included the exchange and dispersal of crop plants, livestock and people[1–9]. This region is also a major geographical and climatic transitional area between the mid-latitude westerlies and the East Asian summer monsoon[10]. Thus, it experienced a complex pattern of temperature variations during the Holocene, which had significant impacts on human migration and crop exchange[11,12], and the related dispersal of domesticated plants and animals. Global and regional palaeoclimate records, regardless of geographical location or temporal resolution, demonstrate substantial climatic variability during the Holocene. Marcott et al.[13] argued for a long-term, cooling trend of Northern Hemisphere surface temperature over the Holocene, while more recent pollen-based climate reconstructions from North America and Europe suggest a long-term warming trend that peaked during ~5.4–4 kyr (1 kyr = 1000 cal yr BP), followed by cooling over

the past ~2000 years[14]. The discrepancy between records of global warming and cooling is called the 'Holocene temperature conundrum'[15]. Model–data inconsistencies are potentially the result of seasonal biases in proxy temperature reconstructions, or model deficiencies[16,17]. More recently, attempts have been made to resolve seasonality signals, deriving annual mean temperature records from seasonal temperature records. The results suggest that global mean annual sea surface temperatures increased continuously since the beginning of the Holocene[18]. However, controversy exists regarding both the magnitude and timing of the Holocene Thermal Maximum (HTM), and the spatial pattern of its onset, duration and magnitude is poorly defined. The HTM is documented in paleoecological records to have occurred during ~9–5 kyr in the Mediterranean region, when mean July temperatures are estimated to have been 1–2 °C warmer than during the recent pre-industrial period[19]. This warming was likely driven by Arctic amplification[17].

[1]Key Laboratory of Western China's Environmental Systems (Ministry of Education), College of Earth and Environmental Sciences, Lanzhou University, 730000 Lanzhou, China. [2]Centre for Environmental Geochemistry, School of Geography, University of Nottingham, Nottingham NG7 2RD, UK. [3]Northwest Institute of Eco-Environmental and Resources, Chinese Academy of Sciences, 730000 Lanzhou, China. [4]Alpine Paleoecology and Human Adaptation Group (ALPHA), Institute of Tibetan Plateau Research, Chinese Academy of Sciences, 100101 Beijing, China. [5]State Key Laboratory of Tibetan Plateau Earth System, Resources and Environment (TPESRE), 100101 Beijing, China. ✉ e-mail: xzhuang@lzu.edu.cn; Virginia.Panizzo@nottingham.ac.uk

Although the long-term warming of mean annual temperatures during the Holocene has been recognized on both global and hemispheric scales[14,18,20], the evolution of temperatures in continental interiors during this period remains controversial. Palynological[21–24] and geochemical data[25,26] from across large areas of northern mid-latitudes provide independent regional comparisons of the temperature variability of the HTM; however, they have yielded divergent or even disparate conclusions. Specifically, records from arid Central Asia (ACA) have shown that Holocene summer temperatures were the highest in the early Holocene[24,27,28], middle Holocene[26], or middle–late Holocene[12,29]. It is important to constrain these regional signals from continental palaeo-records if we are to better understand past and future climatic warming. Additionally, genetic and archeological evidence shows that intensified nomadic pastoralism in the Altai-Sayan region of eastern Eurasia can be traced back to the early Bronze Age, some 5000 years ago[1,30–32]. However, the driving forces and environmental background of this intensification of human activity remain poorly understood.

In this work we analyze the sedimentary archives of Kanas Lake and neighboring Tiewaike Lake, in the Altai Mountains (Fig. 1, Supplementary Notes 1 and 2), with the objective of providing improved temporal constraints on the Holocene climatic variability of this region, where current regional palaeoclimate datasets are contradictory, especially for the middle to late Holocene. Specifically, we analyze the content and origin of the sedimentary organic matter (total organic carbon (TOC), total nitrogen (TN), C/N ratio)), together with geochemical elements, stable isotope ratios ($\delta^{13}C_{org}$, $\delta^{15}N_{org}$, $\delta^{30}Si_{diatom}$), biogenic silica (BSi) and pollen assemblages, to investigate changes in lake primary productivity, and catchment vegetation and chemical weathering, as proxies for changes in local temperature and humidity. We show an exceptionally warm climate during ~6.5–3.6 kyr BP, which is corroborated by other regional climatic records. During 4.7–4.3 kyr BP, especially, a peak in $\delta^{30}Si_{diatom}$ reflects enhanced lake thermal stratification and periodic nutrient limitation as indicated by

concomitant decreasing BSi content. Supported by the widely-recorded humid climate of the middle to late Holocene in arid Central Asia, our geochemical results indicate a significantly warm and wet climate in the Altai Mountain region during 6.5–3.6 kyr, corresponding to the Altai Holocene Climatic Optimum (AHCO). We conclude that the AHCO was critical for promoting an increased prehistoric human population expansion and intensified cultural exchanges across the Central Asian steppe during the Bronze Age.

## Results and discussion

### Definition of the Altai Holocene Climatic Optimum (AHCO)

Forty-six samples for $\delta^{30}Si_{diatom}$ analysis were taken from sediment core KNS15D from Kanas Lake, spanning the past ~14.1 kyr (Fig. 2a). Increases in $\delta^{30}Si_{diatom}$ are caused either by the increased utilization of dissolved silicon (DSi) by diatoms in Kanas Lake (e.g., enhanced productivity) and/or by a decrease of the nutrient supply to the surface water because of reduced convective mixing, or the reduced supply of catchment-derived nutrients (see Supplementary Note 3), and vice versa. BSi concentrations and $\delta^{30}Si_{diatom}$ are indicators of changes in aquatic palaeoproductivity, driven by changes in nutrient availability and/or length of growing season (e.g., summer temperatures)[33,34]. Before 11.7 kyr (Geochemical Zone 1), $\delta^{30}Si_{diatom}$ fluctuated between 0.34 and 0.42‰, accompanied by low BSi (Fig. 2a, b), indicating that the rate of nutrient uptake (especially silicic acid) by lake diatoms was limited by short growing seasons, most likely a result of the low regional temperature. This period was dominated by detrital inputs to Kanas Lake (reflected by sample scores on PC1 of a principal component analysis of the geochemical data), characterized by lithogenic elements (Fig. 2c). At the same time, the sedimentary TOC content, derived mainly from the terrestrial organic influx, was correspondingly low (Fig. 2d), indicating that the regional vegetation cover was low[35] (Fig. 2f). During the onset of the early Holocene (11.7–10.6 kyr, Geochemical Zone 2-1), the high Sr, Zr and coarse grain size (Supplementary Fig. 3) indicate stronger soil erosion, caused by the melting of

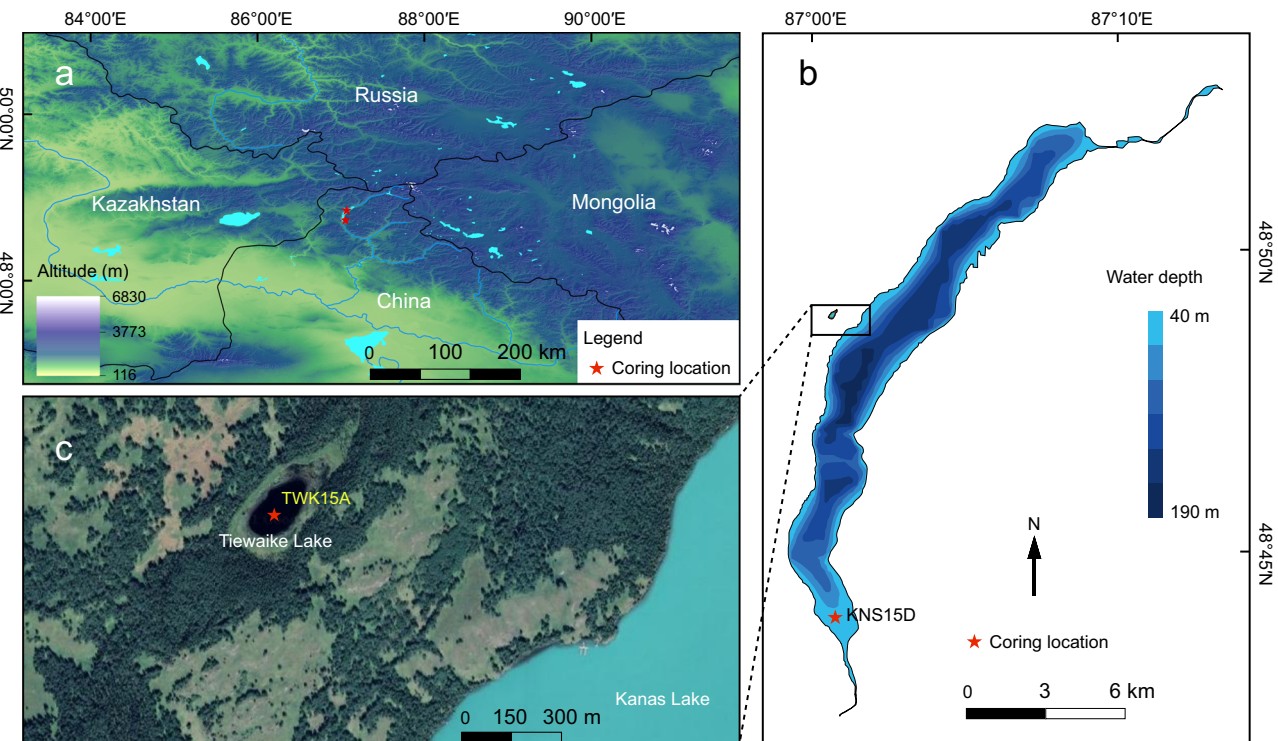

**Fig. 1 | Geographical locations and coring sites of Kanas Lake and neighboring Tiewaike Lake. a** Geographical location of Kanas Lake and Tiewaike Lake in the Altai Mountains. **b** Right panel shows the bathymetry of Kanas Lake (which receives inflowing water from, and outflows to, the Kanas River), together with the coring location. **c** Satellite map of Tiewaike Lake (https://www.earth.google.com/) and the coring location. Coring sites are indicated by red stars.

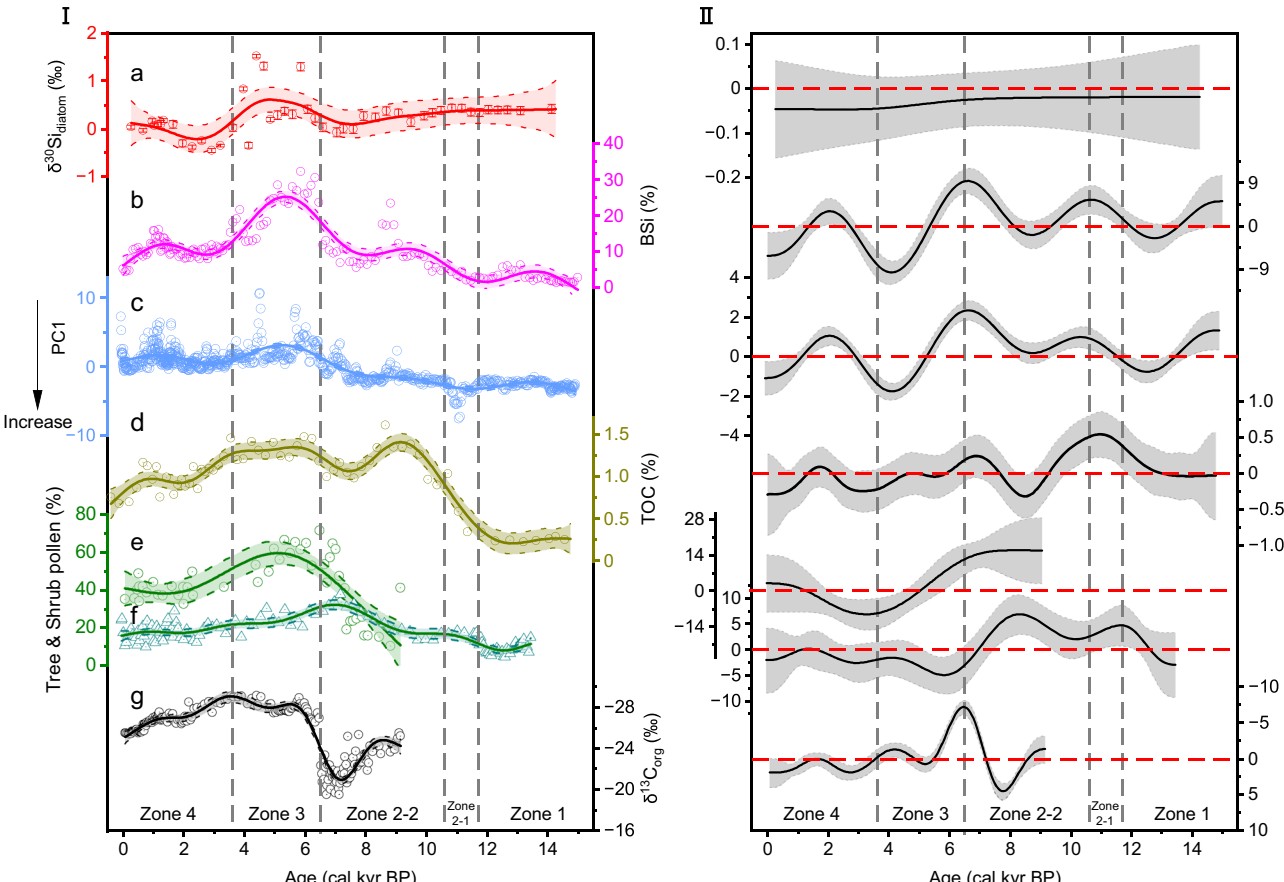

**Fig. 2 | Selected palaeoenvironmental records from Kanas Lake and Tiewaike Lake. I a** $\delta^{30}Si_{diatom}$ with 2-sigma absolute analytical errors, (**b–d**) Biogenic silica, sample scores on PC1 of a principal component analysis (PCA) of the geochemical element data, and the total organic carbon (TOC) content of core KNS15D from Kanas Lake. **e, f** Tree & shrub pollen percentages of core TWK15A from Tiewaike Lake (this study), and core KNS11B from Kanas Lake[35]. **g** $\delta^{13}C_{org}$ from core TWK15A from Tiewaike Lake. Dashed lines correspond to geochemical zones determined by

stratigraphically constrained incremental sum of squares (CONISS) analysis of the scanning X-ray fluorescence measurements. All proxies are fitted with general additive models (GAMs). The shaded bands are 95% confidence intervals. **II** First derivative and the 95% simultaneous confidence interval of the GAMs fitted to each time series (shown in the right panel). Intervals where the derivative deviates from zero represent a significant change in the proxy record. Source data are provided as a Source Data file.

adjacent glaciers combined with a sparse vegetation cover. General additive models (GAMs: see Methods section) were applied to the multiproxy dataset to detect significant temporal changes (at the >95% confidence level), with the lower boundary of Geochemical Zone 2-1 representing a shift to a significantly increased sedimentary TOC content (Fig. 2d) and diatom productivity (Fig. 2b), which together with the PC1 sample scores indicates an increase in the regional biological productivity.

From 10.6 to 6.5 kyr (Geochemical Zone 2-2; see Fig. 2c and Supplementary Note 4), the concentrations of elements indicative of detrital inputs were lower than during the previous zone. During 10.6–8.2 kyr, the significant increase in the sedimentary BSi content (Fig. 2b) indicates the onset of increasing diatom productivity which was accompanied by a trend of decreasing $\delta^{30}Si_{diatom}$ (Fig. 2a), which may have been triggered by an increased silicic acid supply of lower DSi composition, related to intensified catchment weathering, driven by greater vegetation coverage[36] and relatively high summer isolation[37]. These changes were concomitant with a high sedimentary TOC content (Fig. 2d). However, BSi was low during ~8.2–6.5 kyr (Fig. 2b), indicating a decrease in diatom productivity; this is also reflected by the decrease in $\delta^{30}Si_{diatom}$ (Fig. 2a), although it is not highlighted as a significant change by the results of the fitted GAM (not significant at the 95% level). This may have been the result of a dry and possibly cold climate during ~8.2–6.5 kyr, which is regionally corroborated by a low lake level, indicated by the maximum in $\delta^{13}C_{org}$ and

minimum in $\delta^{15}N_{org}$ in the sedimentary record of neighboring Tiewaike Lake (Fig. 2g, Supplementary Fig. 10 and Supplementary Note 5).

The high values of TOC, BSi, and $\delta^{30}Si_{diatom}$ at Kanas Lake, combined with the pollen assemblages and carbon and nitrogen isotopes from Tiewaike Lake, indicate a warm and humid climate during 6.5–3.6 kyr, which defines the Altai Holocene Climatic Optimum (AHCO). High and stable sedimentary TOC concentrations in Kanas Lake (Fig. 2d) corresponded to increasing moisture and vegetation coverage in the Altai Mountains area since ~6.5 kyr[35,38,39]. A significantly lower detrital input, evidenced by PC1 sample scores (Fig. 2c), also occurred between ~6.5 and 3.6 kyr, which can be explained by reduced physical erosion in the catchment, due to the denser vegetation cover which is indicated by higher tree pollen percentages at Tiewaike Lake (Fig. 2e and Supplementary Figs. 9 and 13c). The synchronous changes in $\delta^{30}Si_{diatom}$ and BSi show that both proxies captured the increased silicic acid utilization and diatom productivity during 6.5–3.6 kyr (Fig. 2a, b), with the changes in the BSi record being especially significant (Fig. 2b). The high abundance of *Betula* pollen at Tiewaike Lake (Fig. 2e and Supplementary Figs. 9 and 13c) suggests that the climate was significantly warmer and wetter during 6.5–3.6 kyr, and the pollen record from Kanas Lake also indicates a humid climate at this time (Fig. 2f)[35]. The minimum in $\delta^{13}C_{org}$ and the high values of C/N, Ti, $\delta^{15}N_{org}$ and Rb/Sr between 6.5 and 3.6 kyr suggest the increased contribution of terrestrial plants to the sedimentary OM at high-altitude Tiewaike Lake (Fig. 2g and Supplementary Figs. 10, 11 and 12).

We argue that the $\delta^{30}Si_{diatom}$ record therefore reflects increased diatom productivity driven by higher regional temperatures during 6.5–3.6 kyr, which peaked during 4.7–4.3 kyr, corresponding to the thermal maximum of the AHCO. Indeed, the reduction in the influx of terrigenous material to the lake basin (Fig. 2c), attributed to a denser forest cover and increased soil stability (e.g., as indicated by high *Betula* pollen percentages, Supplementary Figs. 9 and 13c) and a result of the warm and humid climate, led to more prolonged periods of lake thermal stratification with periodic Si limitation contributing to the high values of $\delta^{30}Si_{diatom}$ when BSi declined concomitantly. Additionally, the relatively low values of $\delta^{30}Si_{diatom}$ at 4.2 and 3.6 kyr reflect centennial-scale cooling events, as previously reported at Bosten Lake[12].

### Composite global records of the HTM

A humid HTM climate in ACA is widely recognized (e.g[10,35,40].), and here we focus on the regional expression of the HTM. Based on multi-proxy records, we propose that the AHCO, which accompanied the regional thermal maximum, occurred during 6.5–3.6 kyr and peaked at 4.7–4.3 kyr (Fig. 3). This thermal maximum (Fig. 3a) was simultaneous with the appearance of the thermophilic algal species *Pediastrum simplex* and a high concentration of *Pediastrum duplex* (Chlorophyceae) in Bosten Lake in southern Xinjiang, with warm conditions also indicated by the record of clumped isotopes ($\Delta_{47}$)[12] (Fig. 3b and Supplementary Fig. 8b). Further evidence of a thermal maximum during 6–4 kyr is provided by various independent proxies that reflect environmental changes on a larger spatial scale[41–44]. At lower latitudes, brGDGTs-based temperature records from Lugu Lake (Fig. 3e) and Tengchongqinghai (TCQH) Lake (Fig. 3f), on the southeastern margin of the Tibetan Plateau (TP), show similar changes with a peak during ~4.7–3.6 kyr, while after 3.6–3.5 kyr they indicate a cooling trend[44] (Supplementary Fig. 8c, d). Additionally, multiple brGDGTs-based quantitative temperature reconstructions from several lakes in high mountains in East Africa show that temperatures peaked during 6.5–4.0 kyr[45] (Fig. 3g). In middle- to high-latitude regions, temperature reconstructions based on pollen[46], mollusks[42], and chironomids[41], from Lake Baikal in Russia, northern China, and Alaska, respectively (Fig. 3i), show that summer temperatures and/or growing season temperatures peaked during ~5.5–4.0 kyr. Additionally, a speleothem $\delta^{18}O$ record from the southern Ural Mountains shows a prominent winter temperature peak during ~4.8–4.3 kyr[47] (Fig. 3j).

During the HTM the increased thermal contrast between the ocean and land strengthened the Asian summer monsoon. Pollen-based precipitation reconstructions for northeastern China indicate a Holocene summer monsoon maximum during ~5.5–3.6 kyr, such as at Tianchi Crater Lake[48] (Fig. 3c) and Sihailongwan Maar Lake[21] (Fig. 3d). In addition, an abrupt enhancement of the summer monsoon was documented in northeastern China during 4.6–4.0 kyr, as demonstrated by negative $\delta^{13}C$ values in the Hani peatland[49]. Similarly, the composite $\delta^{18}O$ record for the past ~5.7 kyr from Sahiya Cave, India, shows a peak in monsoon intensity during 4.8–3.8 kyr[50] (Fig. 3h). Possible drivers of the lower temperatures during the early–middle Holocene (11.5–6.3 kyr) were the high frequency of global volcanic eruptions[51] and the melting of Northern Hemisphere ice sheets[15].

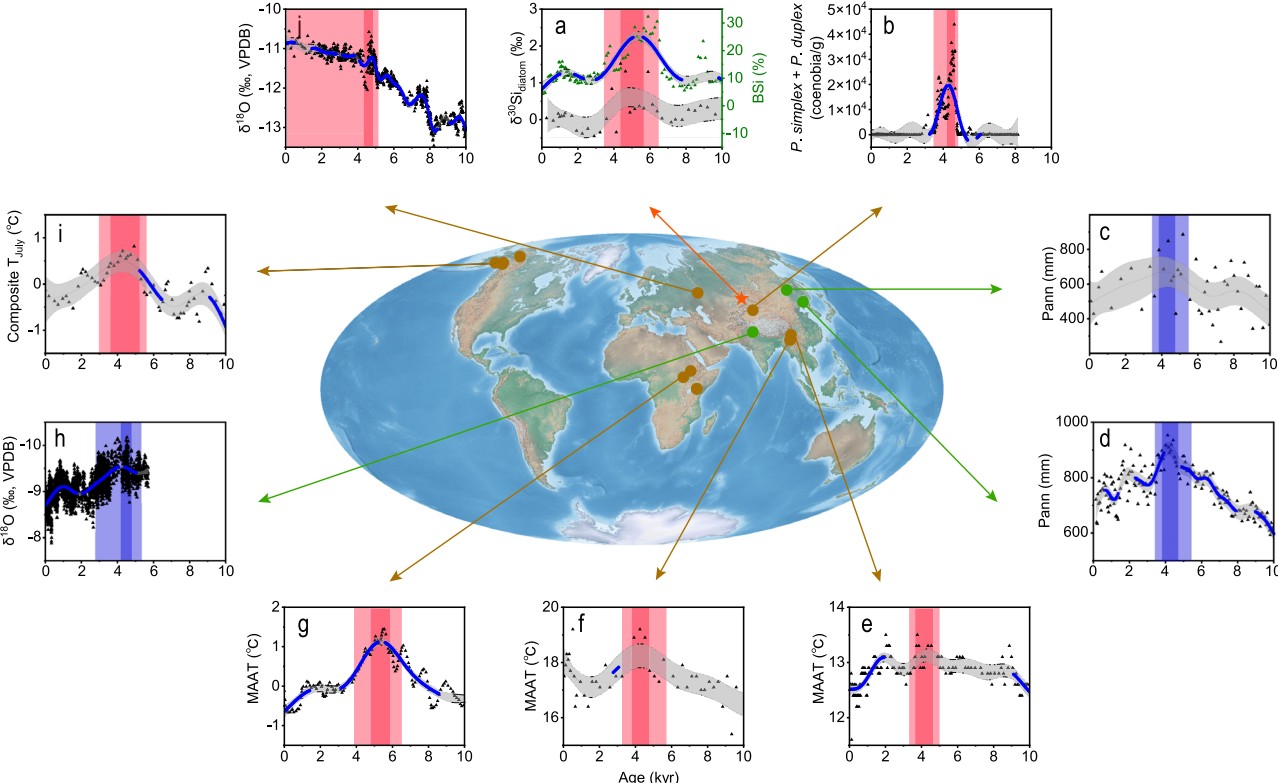

**Fig. 3 | Selected multi-proxy paleoclimate records from various lake and cave sites. a** $\delta^{30}Si_{diatom}$ and BSi records from Kanas Lake (red star). **b** Sum of *P. simplex* and *P. duplex* coenobia from Bosten Lake in Xinjiang[12]. **c, d** Reconstructed Pann from Tianchi Crater Lake[48] and Sihailongwan Lake[21], northeastern China. **e–g** Mean annual air temperature (MAAT) record from Lugu Lake and Tengchongqinghai Lake, on the southeastern margin of the Tibetan Plateau (TP)[44], and a composite brGDGTs record from multiple sites in tropical East Africa[43]. **h** Speleothem $\delta^{18}O$ record from Sahiya Cave, India[50]. **i** Chironomids-based composite summer temperature record from Alaska[41]. **j** Speleothem $\delta^{18}O$ record from Kinderlinskaya Cave, southern Ural Mountains[47]. Brown and green dots on the map indicate temperature and humidity records, respectively. All data are fitted with general additive models (GAMs). The blue dotted line on the GAMs curves identifies significant periods of change for all proxies (i.e., where the derivative deviates significantly from zero). The shaded bands are 95% confidence intervals. The light red/blue shading in the subplots indicates warmer/wetter intervals, while the red/blue bars indicate the warmest/wettest stages. Source data are provided as a Source Data file.

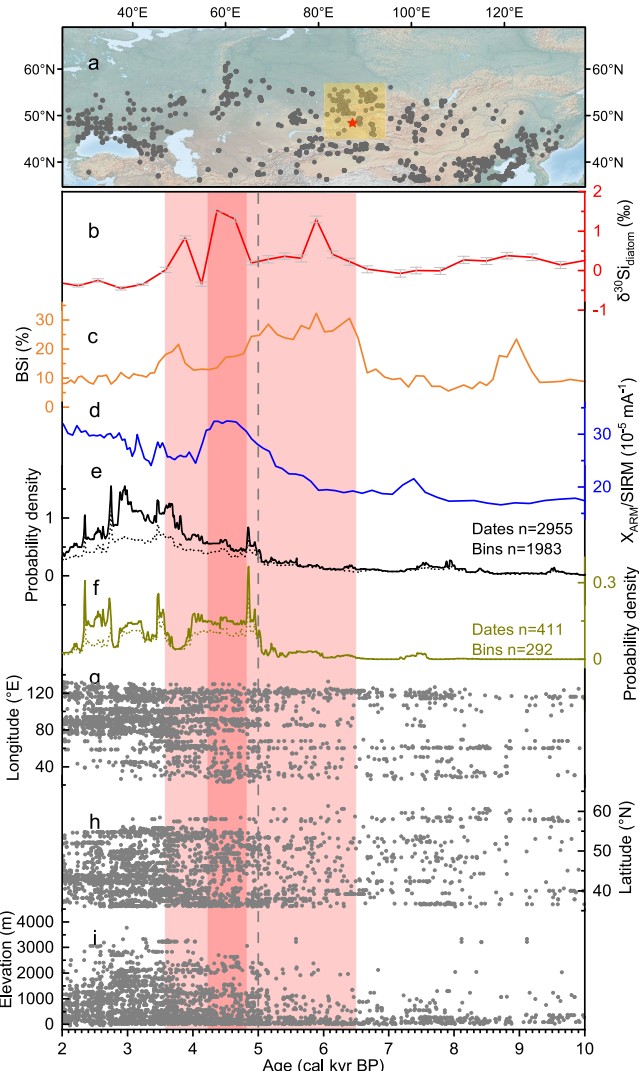

**Fig. 4 | Spatial and temporal distribution of radiocarbon dates from northern Asia and Europe before 2 kyr (0 CE), and comparison with selected palaeoclimatic records. a** Locations of the radiocarbon dates used in this study, and the coring sites (indicated by red stars). **b** Records of δ$^{30}$Si$_{diatom}$ with 2-sigma absolute analytical errors and (**c**) BSi from core KNS15D from Lake Kanas. **d** Environmental magnetic χ$_{ARM}$/SIRM record from the LJW10 loess section in Xinjiang, reflecting pedogenic intensity[40]. Summed probability distribution (SPD) of published original radiocarbon dates from archeological sites before 0 CE in (**e**) this study, and (**f**) the Altai Mountains and the surrounding areas (brown curve, dates are shown in (A) by yellow squares). Dotted line indicates the averaged dates of the bins of the SPD. **g** Longitudinal and temporal distribution of radiocarbon dates. **h** Latitudinal and temporal distribution of radiocarbon dates. **i** Altitudinal and temporal distribution of radiocarbon dates. Archeological radiocarbon dates are available from Dong et al.[76], Huang et al.[12], Taylor et al.[77], and the Canadian Archeological Radiocarbon Database (CARD) at https://www.canadianarchaeology.ca; the references of other dates used in this study are in Supplementary Data 1. The light red shading indicates warmer intervals, while the red bar indicates the warmest stage. Source data are provided as a Source Data file.

## The AHCO promoted human population expansion across the steppe region of Central Asia

The warm and humid climate after 6.5 kyr increased the productivity of mountain grassland, making such areas more attractive to nomadic pastoralists in the northern Eurasian steppe region (Fig. 4a). This is indicated by the increasing summed probability distribution (SPD) of radiocarbon dates of archeological sites (Fig. 4e), a proxy of the size of the human population and settlements (see Methods). The climatic

optimum in the Altai Mountains during ~6.5–3.6 kyr (Fig. 4b–d) may have promoted the expansion of the pastoralist early Bronze Age Afanasievo Culture (~5.1–4.5 kyr) in the Altai-Sayan region[32], and the Chemurchek Culture (~4.5–3.7 kyr) in northern Xinjiang[52] (Fig. 4f). However, a drier climate may have prevailed in the lower basins of western Asia due to the inferred strong evaporation during the HTM; for example, a lake regressive phase is documented during ~5.6–3.7 kyr, shown by a 5–15 m reduction in the water level of the Caspian Sea[53], and the occurrence of a ~600-yr megadrought between ~5.8 and ~5.2 kyr in Kyrgyzstan[54]. Consequently, the Yamnaya Culture (~5.5–4.5 kyr), originating in the Pontic–Caspian region, may have migrated northeastwards into the Afanasievo enclave near the Altai at ~5 kyr, sharing genetic and cultural characteristics[1]. This north-eastward expansion was limited in nature and numerous genetically local groups (ancient North Eurasian (ANE)–derived) persist around this Altai-Sayan region[1,2]. Some of these communities adopted aspects of Afanasievo culture whilst remaining genetically distinct. Concomitantly, there was the more significant westward expansion of the Yamnaya Culture, which also moved into the high Caucasus region during ~4.8–4.3 kyr[55] during this extraordinarily warm period. Both the latitudinal and altitudinal distribution of the numbers of radiocarbon dates from archeological sites during 6.5–3.6 kyr, especially after 5 kyr, show an increasing trend in the Altai-Sayan and surrounding regions (Fig. 4g–i), indicating increased human migration and population expansion in this region under the more favorable environmental conditions.

However, after the pronounced cooling event at ~3.6 kyr, evidence of population movement and/or cultural exchanges is more frequent in oasis-desert areas at lower latitudes[12]. Archeological and genetic evidence shows that the rise of the Xiaohe Culture in southern Xinjiang occurred during ~4.1–3.4 kyr[8,56]. Arguably, these cultural developments were facilitated by the persistent late Holocene wetting trend in ACA[40] (Fig. 4d).

Although previous studies have suggested that the dispersal of agriculture and language[57], domestication of horses[4,30], and dairying[6] may have accompanied and facilitated the dispersal of human populations across Eurasia, our results confirm that the AHCO during 6.5–3.6 kyr, especially during ~4.7–4.3 kyr, provided favorable climatic and ecological conditions for a major phase of prehistoric population expansion and cultural exchange across the steppe region of northern arid Central Asia. This prehistoric population expansion is analogous to the rapid expansion of the Mongolian Empire during the warm and wet climate of the 13th century[58].

## Methods

### Sediment cores, sampling, and chronology

Samples and data were collected independently by our team, following local and national laws. We obtained oral permission from Kanas National Nature Reserve, Xinjiang Province, China to collect samples and data. No official permits were required for this type of sampling, as confirmed by the local authorities. All of the samples were stored in the Key Laboratory of Western China's Environmental Systems (Ministry of Education), College of Earth and Environmental Sciences, Lanzhou University.

In 2015 we sampled the sediments of Kanas Lake (KNS15D, 48°43′19.55″N, 87°1′3.98″E) and Tiewaike Lake (TWK15A, 48°49′36.58N, 87°00′55.54″E), using a UWITEC piston corer, at water depths of 20 m and 5.6 m respectively; the length of each cored sequence was 2.25 m and 5.6 m, respectively (Fig. 1). Core KNS15D was obtained from near the outflow of Kanas Lake. Accelerator mass spectrometry (AMS) $^{14}$C dates from 7 samples of terrestrial plant macrofossils and 3 samples of bulk organic matter (BOM) were obtained from core KNS15D; the analyses were conducted by Beta Analytic Inc. (Florida, USA). Due to the carbon reservoir effect in lacustrine sediments[59], 3 samples of BOM were excluded from the age model (Supplementary Tab.1, Supplementary Fig. 1, and Supplementary Note 2). For Tiewaike Lake, 21 dates

(14 from BOM and 7 from aquatic plant macrofossils) were obtained from core TWK15A; the analyses were conducted by Beta Analytic Inc. and Lanzhou University (Supplementary Tab. 2). The reservoir effect in core TWK15A were assessed in supplementary Note 2 (Supplementary Fig. 14). The AMS [14]C dating results were calibrated to calendar years before present (BP, before 1950 CE) using the Bacon package[60] with the IntCal 20 calibration dataset[61] (Supplementary Fig. 1).

## Geochemical analysis

Element compositions of the sediments were determined by core scanning X-ray fluorescence (XRF-scanning), at a 2-mm resolution, using an Avaatech XRF core scanner[62]. Al, Si, S, Cl, K, Ca, Ti, Mn and Fe were detected at 1 mA and 10 kV for 15 s; and Zn, Rb, Sr and Zr were detected at 2 mA and 30 kV for 25 s. The element data are presented as counts per second (cps). These data are semi-quantitative, reflecting relative changes in chemical composition rather than absolute concentrations. Given that the XRF-scanning results are potentially influenced by water content, surface roughness, and grain-size variations[62,63], the 11 major elements and compositions of 154 samples at a 1-cm resolution from core KNS15D were measured by X-ray Fluorescence Spectrometer (PANalytical B.V., Nederland) using conventional XRF methods at the Key Laboratory of Western China's Environmental Systems, Lanzhou University. Rb, Sr, Zn, Ti, Zr, Mn, $SiO_2$, $Al_2O_3$, $Fe_2O_3$, CaO and $K_2O$, detected by conventional XRF methods, were used in this study (Supplementary Fig. 4). The analytical uncertainties are estimated to be 1–2% for all major metals, and the relative standard deviation is <5% for the trace metals. A correlation analysis of these elements measured by XRF core-scanning and conventional XRF analysis was conducted to assess the validity of the former (Supplementary Figs. 4, 6).

The elements determined by XRF core-scanning (Al, Ti, K, Rb, Zr, Mn, S, Fe, Zn, Cl, Ca, Sr) were analyzed using principal component analysis (PCA) to summarize the variance of the dataset (Supplementary Fig. 5). PCA was performed using R 4.0.5[64]. To overcome the closed-sum effect that leads to spurious correlations between geochemical elements in compositional data, all the XRF data were logarithmically transformed by calculating the central logarithm ratio prior to the PCA[65]. To identify the main factors influencing Rb/Sr in Kanas Lake and Tiewaike Lake, the study examined the correlation between Rb/Sr, Rb, and Sr (Supplementary Note 4, Supplementary Figs. 7, 12). Cluster analysis using stratigraphically constrained incremental sum of squares (CONISS) was used to define geochemical zones based on the elements measured by XRF-scanning.

Diatom silicon isotopes ($\delta^{30}Si_{diatom}$) were measured at the British Geological Survey (BGS), Keyworth, UK. Sample preparation included the removal of contaminants (namely $Al_2O_3$) by vigorous cleaning, including density separation, and oxidation of the organic material (following the methods described in Panizzo et al.[66]). Prior to isotopic analysis, all samples were visually inspected with a Zeiss Axiovert 40C inverted microscope, while XRF analyses were also conducted to quantitatively verify sample purity. All samples demonstrated minimal visual contamination (e.g., by clay) and quantitative estimations via XRF are <1.6% (with the average $Al_2O_3/SiO_2$ ratio of 0.16). Scanning electron microscopy analysis was also conducted to verify that the samples were uncontaminated, for those samples with XRF contamination at the higher threshold scale (e.g., towards 1.6%).

Alkaline fusion (with NaOH) of the cleaned diatom opal and subsequent ion-chromatography (via cation exchange methods; BioRadAG50W-X12) followed the methods in Georg et al.[67]. Samples were analyzed in wet plasma mode using the high mass-resolution capability of a Thermo Scientific Neptune Plus MC-ICP-MS (multi-collector inductively coupled plasma mass spectrometer). A minimum of two analytical replicates were made per sample, with repeated sampling of the standard (diatomite) to validate the data and sample bracketing with standard NBS28 to correct for any instrumental drift (see Panizzo

et al.[66] for further instrumental guidelines). $\delta^{29}Si$ and $\delta^{30}Si$ of diatoms were compared to the mass dependent fractionation line[68] with which all samples comply (Supplementary Fig. 2). Long-term (~2 years) reproducibility and machine accuracy were assessed via analyzing the diatomite secondary standard and the data agree with the published values, as follows. Diatomite: 1.27‰ ± 0.07‰ (2 SD, n = 195) (consensus value of 1.26‰ ± 0.2‰, 2 SD[69]). Biogenic silica (BSi) measurements (Supplementary Fig. 16) were conducted in the Department of Geological Oceanography and State Key Laboratory of Marine Environmental Science, Xiamen University. A complete description of the analytical methods is given in Huang et al.[70].

Samples for analyses of total organic carbon (TOC), total nitrogen (TN) and stable isotope ratios ($\delta^{13}C_{org}$, $\delta^{15}N_{org}$) were pre-treated with 1 N HCl at 60 °C to remove carbonate, washed with deionised water, and freeze-dried prior to analysis. The biogenic silica (BSi) content of the Kanas Lake sediments is relatively high, and therefore to obtain information on the terrestrial sediment source sediment it was necessary to first remove the authigenic sediment component (i.e., diatoms). Therefore, the TOC content of the Kanas Lake sediments was calculated as: TOC% = TOC%×100/(100-BSi%), and the difference between the two methods of TOC calculation is shown in Supplementary Fig. 15. $\delta^{13}C_{org}$ and $\delta^{15}N_{org}$ were measured using an online Conflo III-Delta Plus isotope ratio mass spectrometer combined with a Flash EA1112 elemental analyzer. These results are reported in parts per thousand (‰). $\delta^{13}C_{org}$ and $\delta^{15}N_{org}$ were calculated versus VPDB and atmospheric $N_2$ as the standard, respectively. Replicate analysis of well-mixed samples indicated the precision was better than 0.01‰. TOC and TN percentages were measured using an Elemental Analyzer (VarioEL Cube, Elementar Analysensysteme GmbH, Germany). Grain size analysis was measured at a 1-cm interval for core KNS15D following standard[71]. Grain-size frequency distributions (0.02–2000 μm) were measured using a Malvern Mastersizer 2000 laser grain-size analyzer. These analyses were conducted in Lanzhou University. The palaeo-limnological curves were fitted using generalized additive models (GAMs), using the methods proposed by Simpson[72], to account for the heteroscedasticity typical of this type of data. A smooth GAM model was then used to estimate the trends of the observed time series with restricted maximum likelihood (REML). Periods of statistically significant change were determined by identifying periods when the first derivative of the GAMs deviated from zero[72]. These statistical analyses were conducted using the R packages "mgcv"[73], supplemented with "gratia"[74], based on R 4.0.5 and RStudio 1.2.5001[64].

## Pollen analysis

A total of 49 samples from core TWK15A from Tiewaike Lake were used for pollen analysis, following standard HCl–NaOH–HF treatment[75]. Approximately 1 g of dried sample was used, to which a tablet containing a known number of Lycopodium spores was added to calculate pollen concentrations. Samples were then treated with 10% HCl to remove carbonate, and then with HF to remove siliceous matter. Pollen grains were identified and counted using a Nikon microscope at × 400 magnification; for most samples, at least 500 pollen grains were counted. Pollen samples were stored at Key Laboratory of Western China's Environmental Systems (Ministry of Education), Lanzhou University.

## Prehistoric human activity intensity

The radiocarbon dates were obtained from Dong et al.[76], Huang et al.[12], Taylor et al.[77], and the Canadian Archeological Radiocarbon Database (CARD) at https://www.canadianarchaeology.ca; the references of other published dates used in this study are in Supplementary Data 1. The locational information (e.g., longitude, latitude and altitude) for these sites is mainly from published data, while part of the altitudinal information was extracted from WorldClim 2.1[78], using 30 m Digital Elevation Model (DEM) data. The method of the summed probability distribution (SPD) of radiocarbon dates from archeological

sites is widely applied to reconstruct the variations and intensity of human activity[76,79,80]. The SPDs were generated in OxCal v.4.4.4 using the Sum function[81] at https://c14.arch.ox.ac.uk/, and the dates were calibrated using the IntCal20 calibration curves[61]. To minimize the bias caused by some oversampled sites or site-phases, we processed the dates according to the method proposed by Timpson et al.[82]. Specifically, the radiocarbon dates from the same site where binned using the bin-width of 100 yr, and then an average was obtained by summing the dates within the same bin and dividing by the total number of dates within that bin. Although we acknowledge the limitations of the SPD method used in this study, we consider that the results provide a rough indicator of the intensity of prehistoric human activity.

### Reporting summary
Further information on research design is available in the Nature Portfolio Reporting Summary linked to this article.

## Data availability
The authors declare that all data generated by this study are available within the article and its Supplementary Information/Source Data file/ Supplementary Data. Source data are provided with this paper. The base-maps used in Fig. 1 are accessible through the Geospatial Data Cloud website (https://www.gscloud.cn/) and Google Earth website (https://www.earth.google.com/). The base-maps used in Fig. 3 and Fig. 4 are accessible through the Natural Earth website (https://www.naturalearthdata.com/downloads/10m-raster-data/). Pollen samples used in this study are stored at Key Laboratory of Western China's Environmental Systems (Ministry of Education), Lanzhou University (xzhuang@lzu.edu.cn) and the pollen data are provided in the Source Data file. All data previously published and used in Figs. 3, 4, and Supplementary Fig. 8 can be accessed through the following references[12,21,35,40,41,43,44,47,48,50,76,77]. Source data are provided with this paper.

## Code availability
All of the analyses performed in this study are based on publicly available software programs. Specific version information and non-default arguments are described in the Methods.

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

## Acknowledgements

This work was funded by the National Natural Science Foundation of China (41991251, Huang X.), the National Key Research and Development Program of China (2017YFA0603402, Huang X.), and the National Natural Science Foundation of China (41571182, Huang X.). We thank W. Peng, Z. Bai, L. Wang, Z. Zhu, W. Wang, J. Zhang, X. Ren, and T. Wang for their help with field and laboratory work.

## Author contributions

H.X. designed the study. X.L., H.X., and P.V.N. wrote the manuscript. H.X. and X.L. conducted the fieldwork. S.M. conducted the silicon isotope analyses at the British Geological Survey, UK. X.L. and H.X. conducted part of the laboratory work. H.C., Z.M., C.X., and C.F. and all the others analyzed the data. All authors contributed to this manuscript and approved the final version.

## Competing interests

The authors declare no competing interests.
