## [Peer Review File · Nature Communications]

Prehistoric population expansion in Central Asia promoted by the Altai Holocene Climatic OptimumReviewers' Comments:

Reviewer #1:

Remarks to the Author:

As an archaeological reviewer, I am not sufficiently competent to discuss the climatic reconstructions. However, I can comment on how these are applied to archaeological developments in the region.

Overall this climatic reconstruction will be very valuable to archaeological debates and (if supported by reviewers of climate reconstruction aspects this paper) it is important and these reconstructions need to be published as a matter of priority.

Here are aspects that need to be addressed:

1) Line 250: I think 'promoted' should really be changed to 'may have promoted' so as not to assume causation for correlation.

2) Between lines 257 and 261 the Afanasievo and Yamnaya Cultures are discussed in a disconnected way, but in fact the Afanaseivo enclave near the Altai is a North-easterly expansion of the Yamnaya culture (sharing genetics and cultural characteristics). This NE expansion is limited in nature and many (genetically local, ANE-derived) communities remain around this Afanasievo enclave. Some adopt some aspects of Afanasievo culture whilst being genetically distinct. During the time period discussed for Yamnaya, there is more significant Westward expansion of that culture.

3) Line 261: Some words of caution need to be added around the use of radiocarbon dating. Many sites in the region at this time have not had many radiocarbon dates. Dating has focussed on particular important sites and phenomena where large research budgets have been available. This method will result in a biased impression of settlement density. It is not totally invalid but limitations need to be clear noted otherwise this will be a somewhat naive section of the paper. The paper cites papers that say that this is a reliable indicator, but I have in depth archaeological knowledge of the some of the regions included and know that some sites and site types have been heavily dated for particular research reasons whilst other sites types and periods have been relatively neglected. Sites found more than 20 years ago that have not seen recent work will be particular under-represented. I advise continuing to use this analysis a course indicator but acknowledge a few more limitations.

4) Line 285: Andronovo expansion is discussed, but is should be noted that Andronovo follows on from earlier Sintashta expansion in the Trans-Ural regions of Russian and KZ that starts c. 4.1KYA. Sintashta and Andonovo share admixture of Yamnaya and European farmer genetics that result in human population replacement as well as a change in horse lineage to DOM2 (note: Librado et al. 2021 probably needs citing again for that at line 290).

Reviewer #2:

Remarks to the Author:

The paper studied the Holocene climatic optimum of the Altai Mountains, revealing the warmest and most humid period between ~6.5-3.6 kyr BP with a peak at 4.7-4.3 kyr BP. The undoubted advantage of the study is a relatively rare and reliable method of qualitative reconstruction of temperatures, the measurement of oxygen isotopes in diatom valves.

The discussion of the causes and patterns of climate change in the Holocene of Central Asia in the introduction is somewhat one-sided. The large amount of published data on this region has not been taken into account.

The problem of the climatic optimum in the Altai Mountains and in the region of northern Central Asia as a whole is the focus of numerous studies, which, by the way, are not cited by the authors. For example, the review papers by Zhang, Feng (2018); Klinge, Sauer (2019), which are not cited in the

paper and which reveal a different climate dynamics of the Altai Mountains in the Holocene. Tan et al. (2021) found the presence of a 640-a megadrought between 5820 and 5180 kyr BP, that is, not much earlier than the warmest and more humid interval suggested in this study. Herren et al (2013) revealed that the Tsam-Bagarav glacier in the Mongolian Altai did not begin to grow until the last 6,000 years, thereby marking the neoglacal. By the way, in the cited paper by Huang et al. (2018), both the pollen data and the annual precipitation calculated from them do not show the most humid period between 6.5-3.6 kyr BP. Tree pollen gives a peak at 7 kyr BP and declines further, with precipitation maximal in the late Holocene.

The authors compare the warm and humid period of the Altai Mountains with data from monsoon Asia. I am not sure that this is correct, because different mechanisms in different parts of Asia acted on the Holocene climate (Chen et al., 2008, 2016; Gupta et al., 2003; Kutzbach, 1981; Li et al, 2021 etc). If the authors consider that climate changes in the Altai Mountains and in monsoon Asia were influenced by the same mechanisms, this should be discussed in the paper.

The chronological framework suggested by the authors does not coincide with most of the previously proposed patterns. Authors should discuss and explain why their conclusion, based on the study of two closely located lakes, should be more reliable than all other patterns.

In summary, this paper is a valuable study that will contribute to the development of the understanding of the climate and population of the Altai Mountains. The conclusions and results are based on the application of a set of methods, including a sufficiently rare in climate studies method for measuring oxygen isotopes in diatoms.

To be published in such a prestigious journal, this work must include a discussion of why the scheme proposed by the authors does not correspond to data from other paleoclimate studies of the region. So far, I see that the authors have chosen only those paleoclimate records that support their conclusions.

Specific comments:

The methods should indicate the chronological resolution of the sampling.

I did not see the results of the Tiewaike Lake pollen study, although there was a description in the methods.

Lines 45-47: It cannot be said that the Altai Mountains are a climatic boundary between the westerlies and the Asiatic monsoon. Even in the Early Holocene, according to most paleorecords, Altai was not influenced by the monsoon, although this is also a debatable issue. In the Middle and Late Holocene, the monsoon did not reach the southeastern foothills of the Altai even close.

Lines 117, 160: Mistakes: "Kansas Lake", "vary between 20 and -12‰"

Line 176: Why the birch?

Lines 292-293: The reference is absent: "the Andronovo Culture in Central Asia, during ~3.7–2.5 kyr"

Reviewer #3:

Remarks to the Author:

Key results:

Palaeolimnological investigations of two lakes in the Altai mountains of China are used to show a Holocene thermal Maximum (and increased moisture) occurred in the region between 6.5 – 3.6 kyr BP with a peak in temperatures occurring between 4.7–4.3 kyr. This was demonstrated by the relatively under-utilised $\delta^{30}\text{Si}$ diatom proxy for chemical weathering, productivity, temperature, or water body thermal stratification. This was combined with geochemical proxies for chemical weathering and detrital input (XRF elemental data and ratios), productivity and organic matter provenance ($\delta^{13}\text{C}$, $\delta^{15}\text{N}$, C/N, TOC), climate (pollen), diatom abundance (DSi) and statistical analysis to better constrain changing proxy driver relationships through the Late Pleistocene and Holocene. The authors demonstrate the timing of the Holocene Thermal Maximum and increased moisture in their record, as shown in multiple proxies, is comparable with other records regionally and globally. They also highlight the possible reasons for disparity between their own and some nearby records and cite the complexities of climate in the region. They go on to suggest that the expansion of bronze age

civilisation at around 5 kyr in Central Asia, as recorded by in summed probability distribution of radiocarbon dates of archaeological sites, was facilitated before and during by favourable warmer and wetter conditions recorded at their site.

Validity:

The study uses well established methods to generate an age depth model.

The interpretation of downcore variability in all sediment derived proxies used in this study is difficult without the context of multiple other proxies, however, elemental and ratio geochemistry, $\delta^{30}\text{S}$ diatom, $\delta^{13}\text{C}$, $\delta^{15}\text{N}$, C/N and TN can be especially difficult to interpret individually due to the multiple possible drivers. I feel the authors have made a good effort to ensure that their interpretation is as robust as possible. They have achieved this by taking changes in multiple proxies in context of one another, by referring to appropriate literature and by statistically establishing relationships, for example, through the identification of probable drivers of PCA axis.

Before publication however, I would like to see an explicate interpretation of $\delta^{15}\text{N}$ values in the supplementary material, as has been done with other proxies.

I suspect the authors conclusions could also be better supported by changes the pollen from specific climate indicator species, however the data is not available. I would suggest that the authors include a pollen diagram of the most abundant species in the supplementary information, and if applicable highlight changes in cool/warm or arid/wet indicator species, as they have done with *Betula*.

Significance:

Demonstrates the usefulness of the underutilised $\delta^{30}\text{S}$ diatom.

Adds to the understanding of Holocene climate variability in the mountainous region of Central Asia which, as the authors outline, shows complex and highly variable spatio-temporal climate patterns.

Further demonstrates the significance of climate to the development of early civilization in the region.
Data and methodology

All data included in this study is produced using well established and up to date methods and analytical equipment, and the method is set out concisely. Data errors are duly reported. The presentation of the data is also clear, except for pollen (see earlier comments).

Analytical approach:

The use of generalized additive models is beyond the scope of my expertise. Other analyses are appropriate and are well established in the literature.

Suggested improvements:

As mentioned in previous sections I would like to see an explicate interpretation of $\delta^{15}\text{N}$ values, as has been done so well with other proxies in the method supplementary material.

I would also like to see a pollen diagram of the most abundant species for all samples. I suspect that a focus on certain indicator species of cool/warm or arid/wet conditions will better support the authors findings. I would suggest that the authors include such a pollen diagram in the supplementary information. This will at least support their interpretation of *Betula* (line 156 main text).

I would also like to ask if the authors have considered the role of downcore variability in Mn or Mn/Fe as a proxy for redox process due to lake ventilation/ stratification? See Davison, W., 1993. Iron and manganese in lakes. *Earth-Science Reviews* 34, 119-163, for further details. If not and if appropriate, I would suggest the authors consider Mn and Mn/Fe to further demonstrate their conclusions about

increased lake thermal stratification during the Holocene Thermal Maximum.

I have highlighted suggested minor changes in an annotated PDF of the manuscript.

Clarity and context:

I think the initial issue to be addressed is well researched and laid out in the introduction, the method is clear, the results are fully laid out in the text and figures, and the discussion and conclusions follow a logical order.

References:

With the exception of the following all other references seem appropriate.

Timmermann, A., Yun, K.-S., Raia, P., Ruan, J., Mondanaro, A., Zeller, E., Zollikofer, C., Ponce de León, M., Lemmon, D., Willeit, M., Ganopolski, A., 2022. Climate effects on archaic human habitats and species successions. *Nature* 604, 495–501. - This mainly concerns the Pleistocene. Suggest using a more appropriate reference.

Your expertise:

The use of generalized additive models is beyond the scope of my expertise

Reviewer #4:

Remarks to the Author:

The authors present a multiproxy study about the Holocene climate change and its impact on human migration in the Altai M. Drilling cores from two lakes (Kanas and Tiewaike Lake) in the study area are obtained and the multiproxy including age frames for the cores are all well established. The method for temperature/climate reconstruction ($\delta^{30}\text{Si}$ diatom) is relatively novel, and the paper should be interesting for the readers community for NC. However, I think it still requires revisions before it is accepted for publication.

General comments:

(1) The main problem of the paper is the evolution process of the two studied lakes during the Holocene are not clearly clarified. Because the supplies of the two lakes are different and they should have different evolution process against the same climate condition. From the sedimentary facies (Fig. S8) of Tiewaike Lake, the deposits changed from lacustrine mud to peat (what type of the sediments of the layer "aquatic plant remains"?), it seems that the deep-water lake decreased to a shallow water environment from 9 ka ago. A low Rb/Sr in lake sediments reflects intensified chemical weathering. For Tiewaike Lake, the lowest peak of the Rb/Sr took place during the 8-6.5 ka (Fig. S8B), which means the chemical weathering in the catchment of the lake was the most intensification during that time; and then the catchment of the lake changes to dry from 6.5 ka (Fig. S8B). Or, is there any other reasonable explanation of the lake evolution? The sedimentary facies of Kanas Lake should be the silt in the drilling core (Fig. S3), because the lake is mainly supplied by the glacial melting water, the temperature change maybe the crucial factor to control the lake evolution. The authors argue that the peak in $\delta^{30}\text{Si}$ diatom in Kanas during 6.5-3.6 ka reflects enhanced catchment chemical weathering (high temperature with high precipitation) or increased inputs of glacial meltwater. However, the lowest values of Rb/Sr in Kanas Lake didn't occur during 6.5-3.6 ka (Fig. S3), but during the early Holocene. In addition, the authors haven't provided any direct evidence for the increased inputs of glacial meltwater during this period (the grain size analysis do not show coarser). Please clarify these contradictions.

(2) The results of $\delta^{30}\text{Si}$ diatom of Kanas Lake indicate a high temperature during 6.5-3.6 ka, authors argued that the high precipitation also happened during this period in the study area. However, the tree

& shrub pollen percentages from Tiewaike Lake are vastly different from that of Kanas Lake. The later didn't higher during 6-4 ka but higher during 8-6 ka. This should be explained clearly. With other studies of the climate pattern in the core area of the ACA, the climate type in the study area should be moisture with cold climate or dry with hot climate (Li JY et al., *Climate Dynamics* (2020) 55:1187–1208; Chen, et al, 2016; 2019). The study area has different climate pattern in the westerly dominant core ACA, but similar with the Asian monsoon area, this should be carefully verified.

Specific comments:

(1) The authors state that “ $\delta^{30}\text{Si}$ diatom is a well palaeotemperature proxy in closed lake systems”, and that “ $\delta^{30}\text{Si}$ diatom is often affected by multiple factors, including changes in DSi concentrations and/or compositions supplied by chemical weathering in the catchment, river/aeolian inputs, lake water residence time, changes in stratification/overturning, and other physical characteristics”. Therefore, as an open lake, Kanas Lake probably is not the most suitable target for the $\delta^{30}\text{Si}$ diatom based palaeotemperature reconstruction. For instance, in lines 162-169, the authors can't judge whether the maximum in $\delta^{30}\text{Si}$ diatom during 4.7–4.3 kyr resulted from change in DSi source or chemical weathering. Tiewaike Lake, which has a closed hydrological system, probably is more suitable. However, the authors haven't provided the $\delta^{30}\text{Si}$ diatom results from Tiewaike Lake. The warm climate during ~6.5–3.6 kyr. could be more convincing if the maximum in $\delta^{30}\text{Si}$ diatom can also be tested in Tiewaike Lake. If there is not well-preserved diatom sample could be found in the deposits from Tiewaike Lake, the reason should also be discussed.

(2) The TOC and $\delta^{13}\text{C}$ of Tiewaike can be influenced by the lake-level changes, which should be discussed. For example, $\delta^{13}\text{C}_{\text{org}}$ of Tiewaike Lake reflect the lake from deep water with more aquatic plants (more positive $\delta^{13}\text{C}_{\text{org}}$) to shallow water (peat) with an increase in the proportion of more terrigenous organic matter (more negative $\delta^{13}\text{C}_{\text{org}}$) at 6.5 ka (Fig. S8C). Or, could peat develop in deep water lake as shown in Fig. S9? The process and possible reason of this evolution should be discussed.

(3) It is an important progress if authors could really confirm the rain hot in same period during 6.5–3.6 ka in the Altai M. However, there are many disputes on Holocene temperature change in this area. This result is generally inconsistent with many other temperature records, e.g., the pollen-based temperature anomaly record for 30–90°N (Marcott et al., 2013) and the alkenone-based temperature records from the Balikun Lake (Zhao et al., 2017). What do the authors think of these differences?

Minor comments:

(1) Supplementary Information Line 122-125. When discuss the correlation of $\text{SiO}_2/\text{Al}_2\text{O}_3$ with the BSi, I guess you lost some figures. In Fig. S4, there is not the contents you discussed.

(2) Supplementary Information Line 131. 'Supplementary Fig. S4', it should be S6.

(3) Caption of the Fig. S10, results are from TWK15A from Tiewaike Lake, not from the KNS15D from Kanas Lake.

**Response to Reviewers' Comments on Manuscript NCOMMS-22-47119-T**

**Title:** Prehistoric population expansion in Central Asia promoted by the Altai Holocene Climatic
Optimum

**Reviewer #1 (Remarks to the Author):**

As an archaeological reviewer, I am not sufficiently competent to discuss the climatic
reconstructions. However, I can comment on how these are applied to archaeological
developments in the region.

Overall, this climatic reconstruction will be very valuable to archaeological debates and (if
supported by reviewers of climate reconstruction aspects this paper) it is important and these
reconstructions need to be published as a matter of priority.

---Many thanks for your positive comments.

Here are aspects that need to be addressed:

**Comment 1:** Line 250: I think 'promoted' should really be changed to 'may have promoted' so as
not to assume causation for correlation.

**Reply:** Thanks. Revised. Please see L245 in Main Text.

**Comment 2:** Between lines 257 and 261 the Afanasievo and Yamnaya Cultures are discussed in
a disconnected way, but in fact the Afanaseivo enclave near the Altai is a North-easterly expansion
of the Yamnaya culture (sharing genetics and cultural characteristics). This NE expansion is
limited in nature and many (genetically local, ANE-derived) communities remain around this
Afanasievo enclave. Some adopt some aspects of Afanasievo culture whilst being genetically

distinct. During the time period discussed for Yamnaya, there is more significant Westward
expansion of that culture.

**Reply:** Thank-you for your helpful suggestions. We have added relevant discussion to this effect
in the revision. Please see L253-261 in Main Text.

The Yamnaya Culture (~5.5–4.5 kyr), which originated in the Pontic–Caspian region, may
have migrated north-eastwards into the Afanaseivo enclave near the Altai at ~5 kyr, sharing genetic
and cultural characteristics (Allentoft et al., 2015). This north-eastward expansion was limited in
nature and many genetically local groups (ancient North Eurasian (ANE)-derived) persist around
this Altai-Sayan region (Allentoft et al., 2015; Jeong et al., 2019). Some have adopted certain
aspects of Afanasievo culture while remaining genetically distinct. Also, during this this
extraordinarily warm period, there was the more significant westward expansion of the Yamnaya
Culture which also moved into the high Caucasus region during ~4.8–4.3 kyr (Scott et al., 2022).

References:

Allentoft, M.E., Sikora, M., Sjögren, K.G., Rasmussen, S., Rasmussen, M., Stenderup, J., Damgaard, P.B.,
Schroeder, H., Ahlström, T., Vinner, L., Malaspinas, A.-S., Margaryan, A., Higham, T., Chivall, D.,
Lynnerup, N., Harvig, L., Baron, J., Casa, P.D., Dąbrowski, P., Duffy, P.R., Ebel, A.V., Epimakhov,
40 A., Frei, K., Furmanek, M., Gralak, T., Gromov, A., Gronkiewicz, S., Grupe, G., Hajdu, T., Jarysz, R.,
Khartanovich, V., Khokhlov, A., Kiss, V., Kolář, J., Kriiska, A., Lasak, I., Longhi, C., McGlynn, G.,
Merkevcicius, A., Merkyte, I., Metspalu, M., Mkrtychyan, R., Moiseyev, V., Paja, L., Pálfi, G., Pokutta,
D., Pospieszny, Ł., Price, T.D., Saag, L., Sablin, M., Shishlina, N., Smrčka, V., Soenov, V.I.,
Szeverényi, V., Tóth, G., Trifanova, S.V., Varul, L., Vicze, M., Yepiskoposyan, L., Zhitenev, V.,
Orlando, L., Sicheritz-Pontén, T., Brunak, S., Nielsen, R., Kristiansen, K., Willerslev, E., 2015.
Population genomics of Bronze Age Eurasia. *Nature* 522, 167–172.

Jeong, C., Balanovsky, O., Lukianova, E., Kahbatkyzy, N., Flegontov, P., Zaporozhchenko, V., Immel, A.,
Wang, C.-C., Ixan, O., Khussainova, E., Bekmanov, B., Zaibert, V., Lavryashina, M., Pocheshkhova,
E., Yusupov, Y., Agdzhoyan, A., Koshel, S., Bukin, A., Nymadawa, P., Turdikulova, S., Dalimova,
D., Churnosov, M., Skhalyakho, R., Daragan, D., Bogunov, Y., Bogunova, A., Shtrunov, A., Dubova,
51 N., Zhabagin, M., Yepiskoposyan, L., Churakov, V., Pislegin, N., Damba, L., Saroyants, L., Dibirova,

52 K., Atramentova, L., Utevska, O., Idrisov, E., Kamenshchikova, E., Evseeva, I., Metspalu, M., Outram,
53 A.K., Robbeets, M., Djansugurova, L., Balanovska, E., Schiffels, S., Haak, W., Reich, D., Krause, J.,
2019. The genetic history of admixture across inner Eurasia. *Nat. Ecol. Evol.* 3, 966–976.
Scott, A., Reinhold, S., Hermes, T., Kalmykov, A.A., Belinskiy, A., Buzhilova, A., Berezina, N.,
Kantorovich, A.R., Maslov, V.E., Guliyev, F., Lyonnet, B., Gasimov, P., Jalilov, B., Eminli, J.,
Iskandarov, E., Hammer, E., Nugent, S.E., Hagan, R., Majander, K., Onkamo, P., Nordqvist, K.,
Shishlina, N., Kaverzneva, E., Korolev, A.I., Khokhlov, A.A., Smolyaninov, R.V., Sharapova, S.V.,
Krause, R., Karapetian, M., Stolarczyk, E., Krause, J., Hansen, S., Haak, W., Warinner, C., 2022.
Emergence and intensification of dairying in the Caucasus and Eurasian steppes. *Nat. Ecol. Evol.* 6,
813–822.

**Comment 3:** Line 261: Some words of caution need to be added around the use of radiocarbon
dating. Many sites in the region at this time have not had many radiocarbon dates. Dating has
focused on particular important sites and phenomena where large research budgets have been
available. This method will result in a biased impression of settlement density. It is not totally
invalid but limitations need to be clear noted otherwise this will be a somewhat naive section of
the paper. The paper cites papers that say that this is a reliable indicator, but I have in depth
archaeological knowledge of the some of the regions included and know that some sites and site
types have been heavily dated for particular research reasons whilst other sites types and periods
have been relatively neglected. Sites found more than 20 years ago that have not seen recent work
will be particular under-represented. I advise continuing to use this analysis a course indicator but
acknowledge a few more limitations.

**Reply:** Thank-you for your comments and constructive suggestions. The method of the summed
probability distribution (SPD) of radiocarbon dates from archaeological sites is widely applied to
reconstruct variations in the intensity of human activity (e.g., Wang et al., 2014; Dong et al., 2019;
Briere and Gajewski, 2020). To reduce the bias caused by a few oversampled sites or site-phases,
we refined the dates according to the method propose by Timpson et al. (2014). Specifically, we

binned radiocarbon dates from the same site that were within 100 years, and then averaged the
 dates within that bin. In Fig. 4E-F, the probability density plots of the original radiocarbon dates
 are shown as solid curves, and the probability densities based on the binned and averaged dates
 are shown as dotted curves. Both curves show good agreement (Fig. 4E-F). We acknowledge some
 of the limitations of the SPD method used in this study, but it provides an approximation of the
 intensity of prehistoric human activity. Please see L101-119 in the Methods section.

 Fig. 4. Spatial and temporal distribution of radiocarbon dates from northern Asia and Europe
 before 2 kyr (0 CE), and comparison with selected palaeoclimatic records.

References:

Briere, M.D., Gajewski, K., 2020. Human population dynamics in relation to Holocene climate
variability in the North American Arctic and Subarctic. *Quat. Sci. Rev.* 240, 106370.

Dong, G., Li, R., Lu, M., Zhang, D., James, N., 2020. Evolution of human–environmental
interactions in China from the Late Paleolithic to the Bronze Age. *Prog. Phys. Geogr. Earth
Environ.* 44, 233–250.

Timpson, A., Colledge, S., Crema, E., Edinborough, K., Kerig, T., Manning, K., Thomas, M.G.,
Shennan, S., 2014. Reconstructing regional population fluctuations in the European Neolithic
using radiocarbon dates: a new case-study using an improved method.

Wang, C., Lu, H., Zhang, J., Gu, Z., He, K., 2014. Prehistoric demographic fluctuations in China
inferred from radiocarbon data and their linkage with climate change over the past 50,000
100 years. *Quat. Sci. Rev.* 98, 45–59.

**Comment 4:** Line 285: Andronovo expansion is discussed, but it should be noted that Andronovo
follows on from earlier Sintashta expansion in the Trans-Ural regions of Russian and KZ that starts
c. 4.1 KA. Sintashta and Andonovo share admixture of Yamnaya and European farmer genetics
that result in human population replacement as well as a change in horse lineage to DOM2 (note:
Librado et al. 2021 probably needs citing again for that at line 290).

**Reply:** Thanks. The Andronovo Culture expansion section has been removed, as suggested by
Reviewer #3, because it is outside the time window of the proposed local HTM. The reference
(Librado et al., 2021) has been added. Please see L292 in Main Text.

**Reviewer #2 (Remarks to the Author):**

The paper studied the Holocene climatic optimum of the Altai Mountains, revealing the warmest
and most humid period between ~6.5-3.6 kyr BP with a peak at 4.7-4.3 kyr BP. The undoubted
advantage of the study is a relatively rare and reliable method of qualitative reconstruction of
temperatures, the measurement of oxygen isotopes in diatom valves.

**----Many thanks for your positive comments.**

**Comment 1:** The discussion of the causes and patterns of climate change in the Holocene of
Central Asia in the introduction is somewhat one-sided. The large amount of published data on
this region has not been taken into account.

**Reply:** Thanks for your comments. We agree that the Introduction did not quite reflect the full
breadth of research on ACA in the Holocene, which was due to the length limitation of the
publication. However, we have added further relevant references in the Introduction to address this
comment. In this study, we mainly focus on Holocene temperature reconstruction and discussion.
More details are presented in Reply to Comment 2.

**Comment 2:** The problem of the climatic optimum in the Altai Mountains and in the region of
northern Central Asia as a whole is the focus of numerous studies, which, by the way, are not cited
by the authors. For example, the review papers by Zhang, Feng (2018); Klinge, Sauer (2019),
which are not cited in the paper and which reveal a different climate dynamic of the Altai
Mountains in the Holocene. Tan et al. (2021) found the presence of a 640-a megadrought between
5820 and 5180 kyr BP, that is, not much earlier than the warmest and more humid interval
suggested in this study. Herren et al (2013) revealed that the Tsam-Bagarav glacier in the

Mongolian Altai did not begin to grow until the last 6,000 years, thereby marking the neoglacial.
By the way, in the cited paper by Huang et al. (2018), both the pollen data and the annual
precipitation calculated from them do not show the most humid period between 6.5-3.6 kyr BP.
Tree pollen gives a peak at 7 kyr BP and declines further, with precipitation maximal in the late
Holocene.

**Reply:** Thank-you for your suggestions. We have added a relevant discussion on this and
references. Please see L71-73, L77-78, and 252-253 in Main Text.

The "Altai Holocene climate optimum" as we define it here refers to an optimum combination
of moisture and heat, rather than to a period of highest temperatures and highest precipitation, that
occurred between 6.5 kyr and 3.6 kyr. Our results indicate a warm climate during 6.5–3.6 kyr, with
higher precipitation compared to the early Holocene—which may explain the relatively delayed
timing of human activities at ~5-4 ka. The Tsam-Bagarav glacial advance in the Mongolian Altai
began at ~6 kyr (Herren et al., 2013), which was likely dominated by enhanced precipitation during
the middle to late Holocene. A glacier $\delta^{18}\text{O}$ record from the Russian Altai recorded a warm climate
during ~8–4 kyr, with subsequent cooling (Aizen et al., 2016). Regarding the pollen-based
quantitative precipitation for Kanas Lake (Huang et al., 2018), it should be understood that
pollen/vegetation data combine the signals of both precipitation and temperature, and for large
lakes in high mountains, the large pollen source area with vertical gradients must also be
considered. The reconstruction of Huang et al. (2018) indicates a drier early Holocene and more
humid middle to late Holocene, which may explain why a large Neolithic population had not
previously occupied the study area.

**References:**

Aizen, E.M., Aizen, V.B., Takeuchi, N., Mayewski, P.A., Grigholm, B., Joswiak, D.R., Nikitin,
S.A., Fujita, K., Nakawo, M., Zapf, A., Schwikowski, M., 2016. Abrupt and moderate climate
changes in the mid-latitudes of Asia during the Holocene. *J. Glaciol.* 62, 411–439.
Herren, P.-A., Eichler, A., Machguth, H., Papina, T., Tobler, L., Zapf, A., Schwikowski, M., 2013.
The onset of Neoglaciatioin 6000 years ago in western Mongolia revealed by an ice core from
the Tsambagarav mountain range. *Quat. Sci. Rev.* 69, 59–68.
Huang, X., Peng, W., Rudaya, N., Grimm, E.C., Chen, X., Cao, X., Zhang, J., Pan, X., Liu, S.,
Chen, C., Chen, F., 2018. Holocene Vegetation and Climate Dynamics in the Altai Mountains
and Surrounding Areas. *Geophys. Res. Lett.* 45, 6628–6636.

**Comment 3:** The authors compare the warm and humid period of the Altai Mountains with data
from monsoon Asia. I am not sure that this is correct, because different mechanisms in different
parts of Asia acted on the Holocene climate (Chen et al., 2008, 2016; Gupta et al., 2003; Kutzbach,
1981; Li et al, 2021 etc). If the authors consider that climate changes in the Altai Mountains and
in monsoon Asia were influenced by the same mechanisms, this should be discussed in the paper.

**Reply:** Thanks. We assume that the temperature variations are relatively consistent on a large
spatial scale; however, there may be regional differences in humidity. Here, we only compare
temperature records that are representative of a relatively large spatial scale, while the humidity
records are mainly from Central Asia. In the Asia monsoon regions, there are large regional
differences in precipitation patterns on different timescales (Chen et al., 2015; Huang et al., 2013;
Zhou et al., 2016, 2023). In arid Central Asia (ACA), a generally humid climate is recognized
during the middle to late Holocene (e.g., Chen et al., 2016, 2019; Zhang and Feng, 2018; Huang
et al., 2009, 2018), but there were regional humidity differences between basin and mountainous
areas. High temperatures would enhance the evaporation and result in dry climate in low basin
areas, such as occurred in the Caspian Sea region during the HTM (Bezrodnykh et al., 2020), but
which had a limited impact on mountainous areas. Therefore, in the Altai Mountain area, the

temperature variations were the highest during 6.5–3.6 kyr, with relatively high precipitation at
the same time.

References:

Bezrodnykh, Y., Yanina, T., Sorokin, V., Romanyuk, B., 2020. The Northern Caspian Sea:
Consequences of climate change for level fluctuations during the Holocene. *Quat. Int.* 540,
68–77.

Chen, F., Chen, J., Huang, W., Chen, S., Huang, X., Jin, L., Jia, J., Zhang, X., An, C., Zhang, J.,
Zhao, Y., Yu, Z., Zhang, R., Liu, J., Zhou, A., Feng, S., 2019. Westerlies Asia and monsoonal
Asia: Spatiotemporal differences in climate change and possible mechanisms on decadal to
sub-orbital timescales. *Earth-Sci. Rev.* 192, 337–354.

Chen, F., Jia, J., Chen, J., Li, G., Zhang, X., Xie, H., Xia, D., Huang, W., An, C., 2016. A persistent
Holocene wetting trend in arid central Asia, with wettest conditions in the late Holocene,
revealed by multi-proxy analyses of loess-paleosol sequences in Xinjiang, China. *Quat. Sci.*
*Rev.* 146, 134–146.

Chen, J., Chen, F., Feng, S., Huang, W., Liu, J., Zhou, A., 2015. Hydroclimatic changes in China
and surroundings during the Medieval Climate Anomaly and Little Ice Age: spatial patterns
and possible mechanisms. *Quat. Sci. Rev.* 107, 98–111.

Huang, X., Chen, F., Fan, Y., Yang, M., 2009. Dry late-glacial and early Holocene climate in arid
central Asia indicated by lithological and palynological evidence from Bosten Lake, China.
*Quat. Int.* 194, 19–27.

Huang, X., Peng, W., Rudaya, N., Grimm, E.C., Chen, X., Cao, X., Zhang, J., Pan, X., Liu, S.,
Chen, C., Chen, F., 2018. Holocene Vegetation and Climate Dynamics in the Altai Mountains
and Surrounding Areas. *Geophys. Res. Lett.* 45, 6628–6636.

Huang, X., Xue, J., Wang, X., Meyers, P.A., Huang, J., Xie, S., 2013. Paleoclimate influence on
early diagenesis of plant triterpenes in the Dajiuhu peatland, central China. *Geochim.*
*Cosmochim. Acta* 123, 106–119.

Zhou, X., Zhan, T., Tan, N., Tu, L., Smol, J.P., Jiang, S., Zeng, F., Liu, X., Li, X., Liu, G., Liu, Y.,
Zhang, R., Shen, Y., 2023. Inconsistent patterns of Holocene rainfall changes at the East
Asian monsoon margin compared to the core monsoon region. *Quat. Sci. Rev.* 301, 107952.

Zhou, Xin, Sun, L., Zhan, T., Huang, W., Zhou, Xinying, Hao, Q., Wang, Y., He, X., Zhao, C.,
Zhang, J., Qiao, Y., Ge, J., Yan, P., Yan, Q., Shao, D., Chu, Z., Yang, W., Smol, J.P., 2016.
Time-transgressive onset of the Holocene Optimum in the East Asian monsoon region. *Earth*
*Planet. Sci. Lett.* 456, 39–46.

**Comment 4:** The chronological framework suggested by the authors does not coincide with most
of the previously proposed patterns. Authors should discuss and explain why their conclusion,
based on the study of two closely located lakes, should be more reliable than all other patterns.

**Reply:** We are confident that the chronological framework for this study is reliable. Recently, we
published a paper in *Radiocarbon* about the ^{14}C dating issues of Kanas Lake compared to the
chronologies available for regional sites (Cao et al., 2023). The ^{14}C ages from Kanas Lake are all
based on terrestrial plant remains (e.g., tree twigs, bark, and stem material), which show no obvious
reservoir effect. For the relatively small Lake Tiewaike, 21 AMS ^{14}C dates obtained in this study
and the ^{210}Pb chronology of Li et al. (2017) are used to construct the chronological framework.
The onset of peat development was at ~9–8 kyr, which is like other regional peat sites, such as the
Narenxia Peatland (Feng et al., 2017), Big Black Peatland (Xu et al., 2019) and Tielishahan
Peatland (Zhang et al. 2018). Please see Text S2 in Supplementary Information.

References:

Cao, H., Huang X., Xiang L., 2023. Soil erosion caused the increasing Holocene radiocarbon
reservoir effect of Lake Kanas in the Altai Mountains. *Radiocarbon* 1-14. DOI:
<https://doi.org/10.1017/RDC.2022.93>

Feng, Z., Sun, A., Abdusalih, N., Ran, M., Kurban, A., Lan, B., Zhang, D., Yang, Y., 2017.
Vegetation changes and associated climatic changes in the southern Altai Mountains within
China during the Holocene. *The Holocene* 27, 683–693.

Li, Y., Qiang, M., Zhang, J., Huang, X., Zhou, A., Chen, J., Wang, G., Zhao, Y., 2017.
Hydroclimatic changes over the past 900 years documented by the sediments of Tiewaike
Lake, Altai Mountains, Northwestern China. *Quat. Int.* 452, 91–101.

Xu, H., Zhou, K., Lan, J., Zhang, G., Zhou, X., 2019. Arid Central Asia saw mid-Holocene drought.
*Geology* 47, 255–258.

Zhang, Y., Yang, P., Tong, C., Liu, X., Zhang, Z., Wang, G., Meyers, P.A., 2018. Palynological
record of Holocene vegetation and climate changes in a high-resolution peat profile from the
Xinjiang Altai Mountains, northwestern China. *Quat. Sci. Rev.* 201, 111–123.

In summary, this paper is a valuable study that will contribute to the development of the
understanding of the climate and population of the Altai Mountains. The conclusions and results
are based on the application of a set of methods, including a sufficiently rare in climate studies
method for measuring oxygen isotopes in diatoms.

---Many thanks for your positive comments.

**Comment 5:** To be published in such a prestigious journal, this work must include a discussion of
why the scheme proposed by the authors does not correspond to data from other paleoclimate
studies of the region. So far, I see that the authors have chosen only those paleoclimate records
that support their conclusions.

**Reply:** Thanks for your comment here (including previous comment 2-4) regarding the timings
and nature of main transitions which we put forward in this manuscript. In ACA, Holocene
moisture variations are widely recognized, although the regional differences are possibly related
to differences in the proxies used and the site elevations (Chen et al., 2022; Zhang and Feng, 2018;
Wang and Feng, 2013; Chen et al., 2019; Tian et al., 2022; Liu et al., 2008). Our humidity evidence
is generally in accord with the conclusions of previous studies: that is, humidity increased after
~6.5 kyr at Kanas Lake and Tiewaike Lake.

Indeed, the main contradictions are in the Holocene temperature reconstructions. Here, we
report the new finding that Holocene temperatures peaked during ~4.7–4.3 kyr, with rapid

warming after 6.5 kyr, which is supported by several other records from elsewhere. This point also
helps explain the timing of the intensification of human activities after ~5 kyr.

**References:**

Chen, F., Chen, J., Huang, W., Chen, S., Huang, X., Jin, L., Jia, J., Zhang, X., An, C., Zhang, J.,
Zhao, Y., Yu, Z., Zhang, R., Liu, J., Zhou, A., Feng, S., 2019. Westerlies Asia and monsoonal
Asia: Spatiotemporal differences in climate change and possible mechanisms on decadal to
sub-orbital timescales. *Earth-Sci. Rev.* 192, 337–354.

Chen, S., Chen, J., Lv, F., Liu, X., Huang, W., Wang, T., Liu, J., Hou, J., Chen, F., 2022. Holocene
moisture variations in arid central Asia: Reassessment and reconciliation. *Quat. Sci. Rev.* 297,
107821.

Huang, X., Peng, W., Rudaya, N., Grimm, E.C., Chen, X., Cao, X., Zhang, J., Pan, X., Liu, S.,
Chen, C., Chen, F., 2018. Holocene Vegetation and Climate Dynamics in the Altai Mountains
and Surrounding Areas. *Geophys. Res. Lett.* 45, 6628–6636.

Liu, X., Herzschuh, U., Shen, J., Jiang, Q., Xiao, X., 2008. Holocene environmental and climatic
changes inferred from Wulungu Lake in northern Xinjiang, China. *Quat. Res.* 70, 412–425.

Tian, F., Wang, W., Rudaya, N., Liu, X., Cao, X., 2022. Wet mid–late Holocene in central Asia
supported prehistoric intercontinental cultural communication: Clues from pollen data.
*Catena* 209, 105852.

Zhang, D., Feng, Z., 2018. Holocene climate variations in the Altai Mountains and the surrounding
areas: A synthesis of pollen records. *Earth-Sci. Rev.* 185, 847–869.

**Specific comments:**

The methods should indicate the chronological resolution of the sampling.

**Reply:** Thanks. The temporal sampling resolution is 25–278 yr/cm (average ~98 yr/cm) and ~4-
71 yr/cm (average ~17 yr/cm) for core KNS15D and core TWK15A, respectively. Please see Text
S2 in Supplementary Information.

I did not see the results of the Tiewaike Lake pollen study, although there was a description in the
methods.

**Reply:** Thanks. We have added the pollen diagram for Tiewaike Lake in the Supplementary
Information. Please see Fig. S9.

Lines 45-47: It cannot be said that the Altai Mountains are a climatic boundary between the
westerlies and the Asiatic monsoon. Even in the Early Holocene, according to most paleorecords,
Altai was not influenced by the monsoon, although this is also a debatable issue. In the Middle and
Late Holocene, the monsoon did not reach the southeastern foothills of the Altai even close.

**Reply:** Thanks. Revised. Please see L43-45 in Main Text.

Lines 117, 160: Mistakes: “Kansas Lake”, “vary between 20 and -12%”

**Reply:** Thanks. Revised.

Line 176: Why the birch?

**Reply:** *Betula* mainly grows in the valleys at the lower limit of *Picea* and *Larix* forest (Chen et al.,
2021). *Larix* is the most frost-tolerant tree in the Altai and Siberia (Blyakharchuk and Chernova,
2013). The distribution of temperature-sensitive tree and shrub species from warm to cold is as
follows: *Betula* – *Picea* – *Larix* (Chytrý et al., 2008; Huang et al., 2018). Therefore, in the Kanas
Lake area, *Betula* is regarded as an indicator of a warm, highly humid climate.

References:

Blyakharchuk, T.A., Chernova, N.A., 2013. Vegetation and climate in the Western Sayan Mts
according to pollen data from Lugovoe Mire as a background for prehistoric cultural change
in southern Middle Siberia. *Quat. Sci. Rev.* 75, 22–42.

Chen, L., Li, Y., Zhang, Y., Kong, Z., Qiao, X., Yang, Z., Yan, Q., Zhou, Y., 2021. Relationship
between surface pollen and modern vegetation in northern Xinjiang, China: Implications for
paleovegetation and paleoclimate reconstruction. *Quat. Int.* 589, 124 – 134.

Chytrý, M., Danihelka, J., Kubešová, S., Lustyk, P., Ermakov, N., Hájek, M., Hájková, P., Kočí,
318 M., Otýpková, Z., Roleček, J., Řezníčková, M., Šmarda, P., Valachovič, M., Popov, D., Pišút,
I., 2008. Diversity of forest vegetation across a strong gradient of climatic continentality:
Western Sayan Mountains, southern Siberia. *Plant Ecol.* 196, 61–83.

Huang, X., Peng, W., Rudaya, N., Grimm, E.C., Chen, X., Cao, X., Zhang, J., Pan, X., Liu, S.,
Chen, C., Chen, F., 2018. Holocene Vegetation and Climate Dynamics in the Altai Mountains
and Surrounding Areas. *Geophys. Res. Lett.* 45, 6628–6636.

Lines 292-293: The reference is absent: “the Andronovo Culture in Central Asia, during ~3.7–2.5
326 kyr”

**Reply:** Thanks. The Andronovo Culture expansion section has been removed, as suggested by
Reviewer #3, because it is outside the time window of the proposed local HTM.

**Reviewer #3 (Remarks to the Author):**

Key results:

Palaeolimnological investigations of two lakes in the Altai mountains of China are used to show
a Holocene thermal Maximum (and increased moisture) occurred in the region between 6.5 – 3.6
336 kyr BP with a peak in temperatures occurring between 4.7–4.3 kyr. This was demonstrated by the
337 relatively under-utilised $\delta^{30}\text{Si}_{\text{diatom}}$ proxy for chemical weathering, productivity, temperature, or
338 water body thermal stratification. This was combined with geochemical proxies for chemical
weathering and detrital input (XRF elemental data and ratios), productivity and organic matter

provenance ($\delta^{13}\text{C}$, $\delta^{15}\text{N}$, C/N, TOC), climate (pollen), diatom abundance (DSi) and statistical
analysis to better constrain changing proxy driver relationships through the Late Pleistocene and
Holocene. The authors demonstrate the timing of the Holocene Thermal Maximum and increased
moisture in their record, as shown in multiple proxies, is comparable with other records regionally
and globally. They also highlight the possible reasons for disparity between their own and some
nearby records and cite the complexities of climate in the region. They go on to suggest that the
expansion of bronze age civilisation at around 5 kyr in Central Asia, as recorded by in summed
probability distribution of radiocarbon dates of archaeological sites, was facilitated before and
during by favourable warmer and wetter conditions recorded at their site.

**Reply:** Many thanks for your positive comments.

Validity:

The study uses well established methods to generate an age depth model.

The interpretation of downcore variability in all sediment derived proxies used in this study is
difficult without the context of multiple other proxies, however, elemental and ratio geochemistry,
$\delta^{30}\text{Si}_{\text{diatom}}$, $\delta^{13}\text{C}$, $\delta^{15}\text{N}$, C/N and TN can be especially difficult to interpret individually due to the
multiple possible drivers. I feel the authors have made a good effort to ensure that their
interpretation is as robust as possible. They have achieved this by taking changes in multiple
proxies in context of one another, by referring to appropriate literature and by statistically
establishing relationships, for example, through the identification of probable drivers of PCA axis.
Before publication however, I would like to see an explicate interpretation of $\delta^{15}\text{N}$ values in the
supplementary material, as has been done with other proxies.

I suspect the authors conclusions could also be better supported by changes the pollen from specific
climate indicator species, however the data is not available. I would suggest that the authors
include a pollen diagram of the most abundant species in the supplementary information, and if
applicable highlight changes in cool/warm or arid/wet indicator species, as they have done with
*Betula*.

**Reply:** Thanks for these excellent suggestions that further validate our interpretation of the pattern
of climate change. We have provided detailed replies in the suggested improvements section.

Significance:

Demonstrates the usefulness of the underutilised $\delta^{30}\text{Si}_{\text{diatom}}$. Adds to the understanding of Holocene
climate variability in the mountainous region of Central Asia which, as the authors outline, shows
complex and highly variable spatio-temporal climate patterns. Further demonstrates the
significance of climate to the development of early civilization in the region.

**Reply:** Many thanks for your positive comments.

Data and methodology

All data included in this study is produced using well established and up to date methods and
analytical equipment, and the method is set out concisely. Data errors are duly reported. The
presentation of the data is also clear, except for pollen (see earlier comments).

**Reply:** Many thanks for your positive comments. We have added the pollen diagram for Tiewaike
Lake. Please see Fig. S9 in Supplementary Information.

Analytical approach:

The use of generalized additive models is beyond the scope of my expertise. Other analyses are
appropriate and are well established in the literature.

**Reply:** Many thanks for your positive comments.

Suggested improvements:

**Comment 1:** As mentioned in previous sections I would like to see an explicate interpretation of
$\delta^{15}\text{N}$ values, as has been done so well with other proxies in the method supplementary material.

**Reply:** Oligotrophic alpine/Arctic lakes are often nitrogen limited and they obtain most of their
nitrogen from atmospheric sources such as precipitation and snow melt (Olsen et al., 2013; Wetzel,
2001). The $\delta^{15}\text{N}$ values of the Tiewaike Lake sequence range between -2.2‰ and 0.94‰ (Fig.
S10), which suggests that atmospheric nitrogen ($\delta^{15}\text{N} \sim 0\text{‰}$) is the dominant N source (Talbot and
Johannessen, 1992). During ~8.2-6.5 kyr, the lowest $\delta^{15}\text{N}$ values (from -0.6‰ to -2.2‰, average
397 -1.4‰; Fig. S10) and highest $\delta^{13}\text{C}$ are probably associated with a predominantly aquatic source.
A low lake level and reduced detrital inputs may result in the benthos contributing substantially to
the total lake primary production (Olsen et al., 2013). During 6.5–3.6 kyr, the high $\delta^{15}\text{N}$ values
may result from increased detrital input (terrigenous plant $\delta^{15}\text{N}$ values are 4.5–10.3‰; Talbot and
Johannessen, 1992; Gosling et al., 2022), which increased the $\delta^{15}\text{N}$, suggesting a higher terrigenous
supply. Please see L194-205 in Supplementary Information and 158-161 in Main Text.

References:

Gosling, W.D., Miller, C.S., Shanahan, T.M., Holden, P.B., Overpeck, J.T., van Langevelde, F.,
2022. A stronger role for long-term moisture change than for CO_2 in determining tropical
woody vegetation change. *Science* 376, 653–656.
Olsen, J., Anderson, N.J., Leng, M.J., 2013. Limnological controls on stable isotope records of
late-Holocene palaeoenvironment change in SW Greenland: a paired lake study. *Quat. Sci.*
*Rev.* 66, 85–95.

Talbot, M.R., Johannessen, T., 1992. A high resolution palaeoclimatic record for the last 27 500
411 years in tropical West Africa from the carbon and nitrogen isotopic composition of lacustrine
organic matter. *Earth Planet. Sci. Lett.* 110, 23–37.

Wetzel, R.G., 2001. *Limnology: Lake and River Systems*. Academic Press, San Diego.

**Comment 2:** I would also like to see a pollen diagram of the most abundant species for all samples.
I suspect that a focus on certain indicator species of cool/warm or arid/wet conditions will better
support the authors findings. I would suggest that the authors include such a pollen diagram in the
supplementary information. This will at least support their interpretation of *Betula* (line 156 main
text).

**Reply:** Thank-you for your comments and constructive suggestions. We have added a pollen
diagram from Tiewaike Lake (Fig. S9), and the pollen diagram from Kanas Lake is already
published (Huang et al., 2018). The most abundant taxa in core TWK15A are *Betula* and *Artemisia*.
High *Betula* percentages occurred during ~7–3.6 kyr, indicating a warm and humid climate. The
*Artemisia/Chenopodiaceae* (A/C) shows a long-term increasing trend, suggesting increasing
precipitation since ~7-6 kyr.

Fig. S9. Pollen diagram for core TWK15A from Tiewaike Lake (shading indicates 5 times
 exaggeration of scale).

**References:**

Huang, X., Peng, W., Rudaya, N., Grimm, E.C., Chen, X., Cao, X., Zhang, J., Pan, X., Liu, S.,
 Chen, C., Chen, F., 2018. Holocene Vegetation and Climate Dynamics in the Altai Mountains
 and Surrounding Areas. *Geophys. Res. Lett.* 45, 6628–6636.

**Comment 3:** I would also like to ask if the authors have considered the role of downcore variability
 in Mn or Mn/Fe as a proxy for redox process due to lake ventilation/ stratification? See Davison,
 437 W., 1993. Iron and manganese in lakes. *Earth-Science Reviews* 34, 119-163, for further details. If
 not and if appropriate, I would suggest the authors consider Mn and Mn/Fe to further demonstrate
 their conclusions about increased lake thermal stratification during the Holocene Thermal
 Maximum.

**Reply:** We thank the reviewer for their suggestion. Yes, in the lake environment, hypolimnetic
anoxia can alter the cycling of redox-sensitive elements (e.g., Fe and Mn). Although both Fe and
Mn are soluble under reducing conditions, Mn is usually more soluble than Fe, and thus the Mn/Fe
ratio can be used as a palaeo-redox indicator (Davison et al., 1993; Olsen et al., 2012). Higher
Mn/Fe reflects weaker water stratification (Olsen et al., 2012), a lower lake level, and/or a higher
wind speed (Davies et al., 2015; Haberzettl et al., 2007; Yuan et al., 2022). Based on these
principles, in Tiewaike Lake, an increased Mn/Fe ratio during 8.2–6.5 kyr was caused by oxidizing
conditions together with a lower water level, ignoring possible differences in wind speed (Figs.
S10 and R1). The shift to lower Mn/Fe ratios during 6.5–3.6 kyr may therefore point to a reduction
of the bottom-water oxygen content during periods of enhanced stratification, due to high
temperatures, and/or to de-oxygenation caused by the decomposition of organic material caused
by enhanced biological productivity. Please see L165-176 in Supplementary Information. In
contrast, the minor changes in Mn/Fe (Fig. R1) may not reflect redox variations in Kanas Lake
because it is a large, open and deep lake with a large inflow and a stable lake level.

Fig. R1. Records of Mn, Fe, and Mn/Fe from core TWK15A (A-C) and core KNS15D (D-F).

Fig. S10. Comparison of selected environmental proxies from core TWK15A from Tiewaik Lake.

(A) Ti, (B) Rb/Sr, (C) Mn/Fe, (D) $^{13}\text{C}_{\text{org}}$, (E) $\delta^{15}\text{N}_{\text{org}}$, (F) sum of tree and shrub pollen percentages,

(G) TOC, (H) TN and (I) C/N ratio. The red/gray shading indicates warmer/dryer intervals.

References:

Davies, S.J., Lamb, H.F., Roberts, S.J., 2015. Micro-XRF Core Scanning in Palaeolimnology:
 Recent Developments, in: Croudace, I.W., Rothwell, R.G. (Eds.), *Micro-XRF Studies of*
 *Sediment Cores, Developments in Paleoenvironmental Research*. Springer Netherlands,
 Dordrecht, pp. 189–226.

Davison, W., 1993. Iron and manganese in lakes. *Earth-Sci. Rev.* 34, 119–163.

Haberzettl, T., Corbella, H., Fey, M., Janssen, S., Lücke, A., Mayr, C., Ohlendorf, C., Schäbitz, F.,
 Schleser, G.H., Wille, M., Wulf, S., Zolitschka, B., 2007. Lateglacial and Holocene wet—dry
 cycles in southern Patagonia: chronology, sedimentology and geochemistry of a lacustrine
 record from Laguna Potrok Aike, Argentina. *The Holocene* 17, 297–310.

Olsen, J., Anderson, N.J., Knudsen, M.F., 2012. Variability of the North Atlantic Oscillation over
the past 5,200 years. *Nat. Geosci.* 5, 808–812.

Yuan, K., Sun, Z., Li, C.-G., Ji, K., Hou, X., Wang, M., Hou, J., 2022. Responses of sedimentary
proxy indicators to lake-level fluctuations on the central Tibetan Plateau since the last
deglaciation. *Prog. Phys. Geogr. Earth Environ.* 46, 922–948.

I have highlighted suggested minor changes in an annotated PDF of the manuscript.

**Reply:** Thanks a lot. We have carefully checked and revised the text appropriately.

Clarity and context:

I think the initial issue to be addressed is well researched and laid out in the introduction, the
method is clear, the results are fully laid out in the text and figures, and the discussion and
conclusions follow a logical order.

~~---~~Many thanks for your positive comments.

References:

With the exception of the following all other references seem appropriate.

Timmermann, A., Yun, K.-S., Raia, P., Ruan, J., Mondanaro, A., Zeller, E., Zollikofer, C., Ponce
de León, M., Lemmon, D., Willeit, M., Ganopolski, A., 2022. Climate effects on archaic human
habitats and species successions. *Nature* 604, 495–501. - This mainly concerns the Pleistocene.

Suggest using a more appropriate reference.

**Reply:** Thanks, we have removed this reference.

Your expertise:

The use of generalized additive models is beyond the scope of my expertise.

**Reply:** We have described generalized additive models (GAMs) in the Methods section.

**Reviewer #4 (Remarks to the Author):**

The authors present a multiproxy study about the Holocene climate change and its impact on
human migration in the Altai M. Drilling cores from two lakes (Kanas and Tiewaike Lake) in the
study area are obtained and the multiproxy including age frames for the cores are all well
established. The method for temperature/climate reconstruction ($\delta^{30}\text{Si}_{\text{diatom}}$) is relatively novel, and
the paper should be interesting for the readers community for NC. However, I think it still requires
revisions before it is accepted for publication.

---Many thanks for your positive comments.

General comments:

**Comment 1:** The main problem of the paper is the evolution process of the two studied lakes
during the Holocene are not clearly clarified. Because the supplies of the two lakes are different
and they should have different evolution process against the same climate condition. From the
sedimentary facies (Fig. S8) of Tiewaike Lake, the deposits changed from lacustrine mud to peat
(what type of the sediments of the layer “aquatic plant remains?”), it seems that the deep-water
lake decreased to a shallow water environment from 9 ka ago.

**Reply:** Thank-you very much for your suggestions. We agree that Tiewaike Lake changed from a
deep-water to a shallow-water lake at ~8.2 kyr, as indicated by multiple proxies. We apologize for
the confusion caused by the term “aquatic plant remains”, which we describe as “light brown peat”

in the revision (we have not yet determined its precise composition). According to the relatively
positive $\delta^{13}\text{C}_{\text{org}}$ and sediment lithology during 8.2–6.5 kyr, most of the sediment is comprised of
the “aquatic plant remains”. Please see the photo below (Fig. R2A). Before ~8.2 kyr, the lithology
was gray lacustrine mud (Fig. R2B and R3) with a low organic matter content, and the tree & shrub
pollen percentages, along with total pollen concentrations were low, suggesting a cold climate.
The water level may have been higher than during 8.2–6.5 kyr. During 6.5–3.6 kyr, the more
negative $\delta^{13}\text{C}$ and higher $\delta^{15}\text{N}$ values are associated with increased terrestrial inputs, as further
supported by the Ti content (Fig. S10). The water level would have been higher than during 8.2–
6.5 kyr. We have not yet produced a quantitative lake level reconstruction.

Fig. R2. Photos of aquatic plant remains during 8.2–6.5 kyr (A) and gray lacustrine mud during
9.1–8.2 kyr (B). Note the large difference in composition.

Fig. R3. Photo of core TWK15A showing the lithology change from gray lacustrine mud to

brown peat at ~8.2 kyr.

Fig. S10. Comparison of selected environmental proxies from core TWK15A from Tiewaike
Lake. (A) Ti, (B) Rb/Sr, (C) Mn/Fe, (D) $^{13}\text{C}_{\text{org}}$, (E) $\delta^{15}\text{N}_{\text{org}}$, (F) sum of tree and shrub pollen
percentages, (G) TOC, (H) TN and (I) C/N ratio. The red/gray shading indicates warmer/dryer
intervals.

A low Rb/Sr in lake sediments reflects intensified chemical weathering. For Tiewaike Lake, the
lowest peak of the Rb/Sr took place during the 8-6.5 ka (Fig. S8B), which means the chemical
weathering in the catchment of the lake was the most intensification during that time; and then the
catchment of the lake changes to dry from 6.5 ka (Fig. S8B). Or, is there any other reasonable
explanation of the lake evolution?

**Reply:** In Tiewaike Lake, the Rb/Sr ratios are more strongly correlated with Rb ($r^2=0.49$) than
with Sr ($r^2=0.07$) (Fig. S12). The Rb/Sr ratios are therefore predominantly controlled by the Rb
content, indicating that physical erosion, rather than chemical weathering, was dominant. In
addition, Rb/Sr is strongly correlated with Ti ($r^2=0.88$), and both were low during $\sim 8.2\text{--}6.5$ kyr,
suggesting a lower detrital input during this interval.

Fig. S12. Scatter plots and correlation results for the XRF-scanning data (Sr, Rb, Rb/Sr and Ti)
from core TWK15A, Tiewaike Lake.

Fig. S11. Comparison of selected XRF elements from core TWK15A from Tiewaike Lake. (A)–
 (D) are Ti, Rb, Sr and Rb/Sr, respectively.

The sedimentary facies of Kanas Lake should be the silt in the drilling core (Fig. S3), because the
 lake is mainly supplied by the glacial melting water, the temperature change maybe the crucial
 factor to control the lake evolution. The authors argue that the peak in $\delta^{30}\text{Si}_{\text{diatom}}$ in Kanas during
 6.5-3.6 ka reflects enhanced catchment chemical weathering (high temperature with high
 precipitation) or increased inputs of glacial meltwater.

However, the lowest values of Rb/Sr in Kanas Lake didn't occur during 6.5-3.6 ka (Fig. S3), but
 during the early Holocene.

**Reply:** The implications of the Rb/Sr in Kanas Lake are like those for Tiewaike Lake. The Rb/Sr
ratios are strongly correlated with Rb ($r^2=0.68$) and weakly correlated with Sr ($r^2=0.26$) (Fig. S7),
which indicates active physical erosion and strong detrital inputs, rather than chemical weathering,
within the Kanas Lake catchment. The lowest Rb/Sr ratios in Kanas Lake occurred during 6.5–3.6
571 kyr, indicating lower detrital inputs, as also evidenced by the PC1 sample scores—likely reflecting
the increased vegetation coverage within the Altai Mountains. With reference to the $\delta^{30}\text{Si}_{\text{diatom}}$, the
period before ~ 10.6 to 8.2 kyr is when we argue increased catchment weathering, due to the lower
signatures and concomitant increase in BSi. However, over the period 6.5-3.6 kyr we see the
highest signatures of $\delta^{30}\text{Si}_{\text{diatom}}$, signifying increased nutrient uptake and productivity, indeed with
a period of nutrient limitation (decreased BSi) most likely reflective of increased lake thermal
stratification.

Fig. S7. Scatter plots and correlation results for Rb/Sr and Sr (left) and Rb/Sr and Rb (right) for
Kanas Lake.

In addition, the authors haven't provided any direct evidence for the increased inputs of glacial
meltwater during this period (the grain size analysis do not show coarser). Please clarify these
contradictions.

**Reply:** Sorry for the confusion. The grain size of the Kanas Lake sediments is related to the
geographical conditions within the catchment (Fig. 1). Water-borne particles entering Kanas Lake
are mainly supplied by the Kanas River in the north, and by slope processes. Fine particles are
mainly from Akkul Lake and mountain glaciers, as Kanas Lake is a deep and elongated lake, and
coarse material would be rapidly deposited in the northern part of the basin. The coarse fraction of
the sediments is likely to be mainly from local slope erosion. The Kanas Lake sediments are
generally relatively fine, and the finer particles in the late Holocene likely reflect the higher
contribution of glacial rock flour related to the neoglaciation, while the higher silt content in the
early Holocene is likely derived from local slope processes due to the sparse vegetation cover (and
the smaller contribution of glacial rock flour) (Fig. S3).

Fig. S3. Age profiles of selected geochemical elements determined by XRF-scanning, and sample
 scores on PC2 of a PCA for core KNS15D from Kanas Lake. Geochemical zones determined by
 CONISS are shown on the far right.

**Comment 2:** The results of $\delta^{30}\text{Si}_{\text{diatom}}$ of Kanas Lake indicate a high temperature during 6.5-3.6
 601 ka, authors argued that the high precipitation also happened during this period in the study area.
 However, the tree & shrub pollen percentages from Tiwaike Lake are vastly different from that
 of Kanas Lake. The later didn't higher during 6-4 ka but higher during 8-6 ka. This should be
 explained clearly. With other studies of the climate pattern in the core area of the ACA, the climate
 type in the study area should be moisture with cold climate or dry with hot climate (Li JY et al.,
 Climate Dynamics (2020) 55:1187–1208; Chen, et al, 2016; 2019). The study area has different

climate pattern in the westerly dominant core ACA, but similar with the Asian monsoon area, this
should be carefully verified.

**Reply:** Thanks for your comments. (1) Tiewaike Lake is a small, closed-basin lake, with no stream
inputs or outputs, with the surface area of only ~ 0.02 km². The pollen source area is limited to the
local environment. However, Kanas Lake, it is a large, open lake with an inflowing river, and the
pollen source area is large and spans a wide altitudinal range including mountain slopes and basins.
The tree & shrub pollen percentages in core TWK15A are higher than those in core KNS15B (Fig.
R4), which is in accord with its presumed large pollen source area in Kanas Lake. Multi-proxy
data from Tiewaike Lake show that a low lake level and the increased representation of aquatic
plants occurred during 8.2–6.5 kyr, which may be the reason for the relatively low tree & shrub
pollen representation during this interval.

(2) We have also clarified this issue in our response to Reviewer #2. We assume that the
temperature variations are relatively consistent on a large spatial scale; however, the humidity may
have varied on a regional scale. Here, we are comparing temperature records on a relatively large
spatial scale and the humidity records are mainly from Central Asia. In the Asian monsoon regions,
there are large regional differences in precipitation patterns on different timescales (Chen et al.,
2015; Huang et al., 2013; Zhou et al., 2016, 2023). In ACA, a generally humid climate is widely
recognized during the middle to late Holocene (e.g., Chen et al., 2016, 2019; Zhang and Feng,
2018; Huang et al., 2009, 2018), but humidity differences existed between basin and mountainous
areas. High temperatures would enhance the evaporation and result in dry climate in basin areas,
as in the case of the dry climate in the Caspian Sea region during the HTM (Bezrodnykh et al.,
2020), although this had a limited impact on mountainous areas. Therefore, in the Altai Mountains,

temperatures were the highest during 6.5–3.6 kyr, and relatively high precipitation also occurred
during this interval.

References:

Bezrodnykh, Y., Yanina, T., Sorokin, V., Romanyuk, B., 2020. The Northern Caspian Sea:
Consequences of climate change for level fluctuations during the Holocene. *Quat. Int.* 540,
68–77.

Chen, F., Chen, J., Huang, W., Chen, S., Huang, X., Jin, L., Jia, J., Zhang, X., An, C., Zhang, J.,
Zhao, Y., Yu, Z., Zhang, R., Liu, J., Zhou, A., Feng, S., 2019. Westerlies Asia and monsoonal
Asia: Spatiotemporal differences in climate change and possible mechanisms on decadal to
sub-orbital timescales. *Earth-Sci. Rev.* 192, 337–354.

Chen, F., Jia, J., Chen, J., Li, G., Zhang, X., Xie, H., Xia, D., Huang, W., An, C., 2016. A persistent
Holocene wetting trend in arid central Asia, with wettest conditions in the late Holocene,
revealed by multi-proxy analyses of loess-paleosol sequences in Xinjiang, China. *Quat. Sci.*
*Rev.* 146, 134–146.

Chen, J., Chen, F., Feng, S., Huang, W., Liu, J., Zhou, A., 2015. Hydroclimatic changes in China
and surroundings during the Medieval Climate Anomaly and Little Ice Age: spatial patterns
and possible mechanisms. *Quat. Sci. Rev.* 107, 98–111.

Huang, X., Chen, F., Fan, Y., Yang, M., 2009. Dry late-glacial and early Holocene climate in arid
central Asia indicated by lithological and palynological evidence from Bosten Lake, China.
*Quat. Int.* 194, 19–27.

Huang, X., Peng, W., Rudaya, N., Grimm, E.C., Chen, X., Cao, X., Zhang, J., Pan, X., Liu, S.,
Chen, C., Chen, F., 2018. Holocene Vegetation and Climate Dynamics in the Altai Mountains
and Surrounding Areas. *Geophys. Res. Lett.* 45, 6628–6636.

Huang, X., Xue, J., Wang, X., Meyers, P.A., Huang, J., Xie, S., 2013. Paleoclimate influence on
early diagenesis of plant triterpenes in the Dajiuhu peatland, central China. *Geochim.*
*Cosmochim. Acta* 123, 106–119.

Zhou, X., Zhan, T., Tan, N., Tu, L., Smol, J.P., Jiang, S., Zeng, F., Liu, X., Li, X., Liu, G., Liu, Y.,
Zhang, R., Shen, Y., 2023. Inconsistent patterns of Holocene rainfall changes at the East
Asian monsoon margin compared to the core monsoon region. *Quat. Sci. Rev.* 301, 107952.

Zhou, Xin, Sun, L., Zhan, T., Huang, W., Zhou, Xinying, Hao, Q., Wang, Y., He, X., Zhao, C.,
Zhang, J., Qiao, Y., Ge, J., Yan, P., Yan, Q., Shao, D., Chu, Z., Yang, W., Smol, J.P., 2016.
Time-transgressive onset of the Holocene Optimum in the East Asian monsoon region. *Earth*
*Planet. Sci. Lett.* 456, 39–46.

Fig. R4. Tree and shrub pollen percentages from Tiewaike Lake and Kanas Lake.

Specific comments:

(1) The authors state that “ $\delta^{30}\text{Si}_{\text{diatom}}$ is a well palaeotemperature proxy in closed lake systems”,
and that “ $\delta^{30}\text{Si}_{\text{diatom}}$ is often affected by multiple factors, including changes in DSi concentrations
and/or compositions supplied by chemical weathering in the catchment, river/aeolian inputs, lake
water residence time, changes in stratification/overturning, and other physical characteristics”.
Therefore, as an open lake, Kanas Lake probably is not the most suitable target for the $\delta^{30}\text{Si}_{\text{diatom}}$
based palaeotemperature reconstruction. For instance, in lines 162-169, the authors can’t judge

whether the maximum in $\delta^{30}\text{Si}_{\text{diatom}}$ during 4.7–4.3 kyr resulted from change in DSi source or
chemical weathering. Tiewaike Lake, which has a closed hydrological system, probably is more
suitable. However, the authors haven't provided the $\delta^{30}\text{Si}_{\text{diatom}}$ results from Tiewaike Lake. The
warm climate during ~6.5–3.6 kyr. could be more convincing if the maximum in $\delta^{30}\text{Si}_{\text{diatom}}$ can
also be tested in Tiewaike Lake. If there is not well-preserved diatom sample could be found in
the deposits from Tiewaile Lake, the reason should also be discussed.

**Reply:** Thanks for your comments. $\delta^{30}\text{Si}_{\text{diatom}}$ has been shown to be a reliable proxy in large, open
lake systems (e.g., Lake Baikal), in which the diatoms are very well preserved, as is in the case of
Kanas Lake. Especially also in lakes with strong seasonal ice cover as their bloom period will be
well constrained by the period of ice off, thermal warming of the water column and nutrient
availability. Therefore, acting as a proxy for summer water temperatures (e.g., Lake Baikal
(Panizzo et al., 2018) and Lake El'gygytgyn (Swann et al., 2010)). The sediments of Tiewaike
Lake are organic-rich and some clayey sediments also occur. The lake water is relatively rich in
humic acids derived from the peaty forest soils in the catchment. Prior to 3.6 kyr, BSi was relatively
low (<5%) (Fig. S16), and the sedimentary organic matter content was high. The biogenic silica
may also be influenced by the peaty lake water, in addition to climate change. The diatom
concentrations in this lake are low and there is a large proportion of Chrysophytes. This would
make the potential to isolate the two forms of biogenic silica very difficult via heavy density
separation and/or sieving. Not being able to do this effectively would have adversely affect the
$\delta^{30}\text{Si}_{\text{diatom}}$ record. Furthermore, when we analysed the biogenic silica we found the lowest values,
together with the almost complete absence of diatoms, occurred before ~6.5 kyr. During ~6.5–3.6
694 kyr, BSi fluctuated within the range of ~2.5–4.0%. After 3.6 kyr, BSi at Tiewaike Lake increased,
while the opposite trend occurred at Kanas Lake. This suggests that the while the neoglaciation in

the Altai Mountains significantly influenced Kanas Lake, it had little impact on the small closed-
basin of Tiewaike Lake. Please see Text S3 in Supplementary Information.

Fig. S16. Biogenic silica profile for core TWK15A from Tiewaike Lake.

References:

Panizzo, V.N., Swann, G.E.A., Mackay, A.W., Pashley, V., Horstwood, M.S.A., 2018. Modelling
silicon supply during the Last Interglacial (MIS 5e) at Lake Baikal. *Quat. Sci. Rev.* 190, 114–
122.

Swann, G.E.A., Leng, M.J., Juschus, O., Melles, M., Brigham-Grette, J., Sloane, H.J., 2010. A
combined oxygen and silicon diatom isotope record of Late Quaternary change in Lake
El'gygytyn, North East Siberia. *Quat. Sci. Rev.* 29, 774–786.

(2) The TOC and $\delta^{13}\text{C}$ of Tiewaike can be influenced by the lake-level changes, which should be
discussed. For example, $\delta^{13}\text{C}_{\text{org}}$ of Tiewaike Lake reflect the lake from deep water with more
aquatic plants (more positive $\delta^{13}\text{C}_{\text{org}}$) to shallow water (peat) with an increase in the proportion of

more terrigenous organic matter (more negative $^{13}\text{C}_{\text{org}}$) at 6.5 ka (Fig. S8C). Or, could peat develop
in deep water lake as shown in Fig. S9? The process and possible reason of this evolution should
be discussed.

**Reply:** Thanks. The peat developed at ~ 8.2 kyr when there was an increased proportion of
sedimentary organic matter derived from aquatic plants, while at the same time there was a
decrease in detrital inputs due to the dry climate and lower precipitation. During 6.5–3.6 kyr, more
negative $\delta^{13}\text{C}_{\text{org}}$ and high sedimentary Ti indicate an increase in terrestrial inputs caused by the
enhanced precipitation. Please see Supplementary Materials for a detailed discussion.

(3) It is an important progress if authors could really confirm the rain hot in same period during
6.5-3.6 ka in the Altai M. However, there are many disputes on Holocene temperature change in
this area. This result is generally inconsistent with many other temperature records, e.g., the pollen-
based temperature anomaly record for 30–90°N (Marcott et al., 2013) and the alkenone-based
temperature records from the Balikun Lake (Zhao et al., 2017). What do the authors think of these
differences?

**Reply:** Many thanks for your comments. There is increasing evidence supporting our arguments
for the onset of high temperature and high rainfall at ~ 6.5 kyr which peaked at 4.7-4.3 kyr, with
short cooling events at 4.2 and 3.6 kyr. Marcott et al. (2013) synthesized 73 records that showed
the temperature maximum at ~ 7 kyr with pronounced cooling thereafter. However, these records
are mostly from marine and coastal settings which have a seasonal bias (Liu et al., 2014; Marsicek
et al., 2018). Pollen-based climate reconstructions from North America and Europe suggest a long-
term warming trend that peaked during ~ 5.4 –4 kyr (Marsicek et al., 2018).

Balikun Lake is a closed-basin, hyper-saline lake (salinity ~94–126 g/L) at the present-day,
and the alkenone proxy may have been affected by salinity. There are also uncertainties regarding
the chronology of Balikun Lake. Although the carbon reservoir was assessed, there are only four
radiocarbon ages for the entire Holocene (Zhao et al., 2017). In addition, the reconstructed
temperature was highest during 8–4 kyr, which overlaps with the interval of in our study (6.5–3.6
739 kyr).

References:

Liu, Z., Zhu, J., Rosenthal, Y., Zhang, X., Otto-Bliesner, B.L., Timmermann, A., Smith, R.S.,
Lohmann, G., Zheng, W., Elison Timm, O., 2014. The Holocene temperature conundrum.
Proc. Natl. Acad. Sci. 111, E3501–E3505.

Marcott, S.A., Shakun, J.D., Clark, P.U., Mix, A.C., 2013. A Reconstruction of Regional and Global
Temperature for the Past 11,300 Years. Science 339, 1198–1201.

Marsicek, J., Shuman, B.N., Bartlein, P.J., Shafer, S.L., Brewer, S., 2018. Reconciling divergent trends and
millennial variations in Holocene temperatures. Nature 554, 92–96.

Zhao, J., An, C.B., Huang, Y., Morrill, C., Chen, F.H., 2017. Contrasting early Holocene temperature
variations between monsoonal East Asia and westerly dominated Central Asia. Quat. Sci. Rev. 178,
14–23.

Minor comments:

(1) Supplementary Information Line 122-125. When discuss the correlation of $\text{SiO}_2/\text{Al}_2\text{O}_3$ with
the BSi, I guess you lost some figures. In Fig. S4, there is not the contents you discussed.

**Reply:** Thanks. We have included them. Please see Fig. S6.

Fig. S6. Scatter plots and correlation results for the relationships between SiO₂/Al₃O₂, SiO₂ and

BSi.

(2) Supplementary Information Line 131. ‘Supplementary Fig. S4’, it is should be S6.

**Reply:** Thanks. Revised.

(3) Caption of the Fig. S10, results are from TWK15A from Tiewaike Lake, not from the KNS15D

from Kanas Lake.

**Reply:** Thanks. Revised.

Reviewers' Comments:

Reviewer #1:

Remarks to the Author:

In this revised version the authors have appropriately responded to my comments. I am fully satisfied and recommend publication of this version.

Reviewer #2:

Remarks to the Author:

As this is a second review of the paper by Xiang et al. "Prehistoric population expansion in Central Asia promoted by the Altai Holocene Climatic Optimum", I will only focus on the authors' responses and corrections, in line with my comments.

Despite the fact that the authors have done a great job in correcting the paper, I still did not see convincing evidence in the discussion that the climate scheme proposed by the authors is more correct than all the others. It is not clear what should be done about glacial retreat. For example, in the Russian Altai, very close to Lake Kanas, the Akkem stage of glacial advance has been dated to 4.9-4.2 ka BP (Agatova et al., 2021). The authors also continue to compare the climate of the northwestern Altai (in the sense of the Big Altai) with that of monsoonal Central Asia, rather than, for example, with the palaeo-records of southwestern Siberia, the Russian Altai or the Mongolian Altai. Although it has to be admitted that such palaeorecords are few and far between, they do exist.

This repetition of mine does not detract from the importance of the study, but it does impoverish the discussion somewhat. However, the authors have responded to my comments, mostly in Response to Reviewers, and therefore this view of the discussion can be seen as the authors' view.

If the other reviewers have no objections, my recommendation is to accept the paper for publication.

Of the minor errors, I could not find the reference to Cao et al, 2023, in the Supplementary.

Fig. S16 shows the biogenic silica profile not only from Lake Tiewaike but also from Lake Kanas.

The word pollen is misspelt in the table - polle

Reviewer 3's response to the author's rebuttal.

Thank you very much for taking the time to respond to my remarks. I hope you found them useful. I am happy that you have addressed all of my minor suggestions. Below are my responses to comments 1 -3.

Comment 1 ($\delta^{15}\text{N}$):

Thank you, you have adequately addressed my comment.

Comment 2 (Pollen):

I am happy with the pollen diagram, but I would suggest including the meaning A/C in the figure caption.

I also suggest that you include some text in the discussion that highlights the significance of *Artemisia* and *Chenopodiaceae* data and how it fits with your over all story.

Comment 3 (Mn/Fe):

Thanks for adding the extra graphics and text. I agree with your discussion and have some minor additions for the text highlighted in red below.

In the lake environment, hypolimnetic anoxia can alter the cycling of redox-sensitive 170 elements. Although both Fe and Mn are soluble under reducing conditions, Mn is usually 171 more soluble than Fe, and thus the Mn/Fe ratio can be used as a palaeo-redox proxy (Davies 172 et al., 1993; Olsen et al., 2012). Higher Mn/Fe ratios mainly reflect weaker stratification 173 (Olsen et al., 2012), as a result of, for example, lower water level and/or higher wind speed (Davies et al., 2015; 174 Haberzettl et al., 2007; Yuan et al., 2022). In Tiewaike Lake, the increased Mn/Fe ratios 175 (Fig. S10) may reflect the predominantly oxidizing lake status at depth during 8.2–6.5 kyr, together 176 with a lower water level, ignoring the possible effect of changes in wind speed. A shift to 177 lower Mn/Fe ratios may therefore point to a lowering of the oxygen content of the bottom 178 water during enhanced stratification due to high temperatures, and/or to deoxygenation 179 caused by the decomposition of organic matter following the enhanced biological 180 productivity during 6.5–3.6 kyr.

Thank you again and good luck with the manuscript.

Reviewer #4:

Remarks to the Author:

The authors have carefully considered the comments, and present new data/figures and discussions to support their conclusions. Overall I think the data and discussion are more convincing and robust. I have some additional minor comments, which should be addressed easily.

1. In Supplementary Information Text, Line 44, please provide the depth of the Tiewaike Lake.
2. The Tiewaike Lake is mainly recharged by the precipitation and keeps several meters depth fresh water since ~6.5 ka ago. It is interesting that why a lake with closed lake basin could keep fresh for 6.5 ka; and why its lacustrine sediments are peat like deposits since 8.2 ka ago (Fig. R3). Please clarify these in supplementary or in main text when discuss the Figure S13.

Response to Reviewers' Comments on Manuscript NCOMMS-22-47119A

Title: Prehistoric population expansion in Central Asia promoted by the Altai Holocene Climatic Optimum

REVIEWERS' COMMENTS

Reviewer #1 (Remarks to the Author):

In this revised version the authors have appropriately responded to my comments. I am fully satisfied and recommend publication of this version.

Reply: Many thanks for your positive evaluation of this manuscript and your support.

Reviewer #2 (Remarks to the Author):

As this is a second review of the paper by Xiang et al. "Prehistoric population expansion in Central Asia promoted by the Altai Holocene Climatic Optimum", I will only focus on the authors' responses and corrections, in line with my comments. Despite the fact that the authors have done a great job in correcting the paper, I still did not see convincing evidence in the discussion that the climate scheme proposed by the authors is more correct than all the others. It is not clear what should be done about glacial retreat. For example, in the Russian Altai, very close to Lake Kanas, the Akkem stage of glacial advance has been dated to 4.9-4.2 ka BP (Agatova et al., 2021). The authors also continue to compare the climate of the northwestern Altai (in the sense of the Big Altai) with that of monsoonal Central Asia, rather than, for example, with the palaeo-records of southwestern Siberia, the Russian Altai or the Mongolian Altai. Although it has to be admitted that such palaeorecords are few and far between, they do exist.

This repetition of mine does not detract from the importance of the study, but it does impoverish the discussion somewhat. However, the authors have responded to my comments, mostly in Response to Reviewers, and therefore this view of the discussion can be seen as the authors' view. If the other reviewers have no objections, my recommendation is to accept the paper for publication.

Reply: Thanks for your comments. The analysis of new chronological data from the Russian Altai suggests that there was lesser glacial activity between 5 and 4 kyr, which is also supported by rapid reforestation in the heads of trough valleys (Agatova et al., 2021). We propose that a small-scale glacier advance was caused by an increase in precipitation due to enhanced moisture during the 5-4 kyr in Russia's Altai region (Rudaya et al., 2020), rather than lower temperature. Furthermore, a quantitative reconstruction of summer temperatures based on pollen data from south-western Siberia indicates that summer temperatures were higher than present-day levels between 4.9 and 4.2 kyr, and subsequently decreased during the late Holocene (Zhilich et al., 2017). We hope that this elucidates our position of the discussion. Thank you.

References:

- Agatova, A., Nepop, R., Nazarov, A., Ovchinnikov, I., Moska, P., 2021. Climatically driven Holocene glacier advances in the Russian Altai based on radiocarbon and OSL dating and tree ring analysis. *Climate*, 9, 162.
- Rudaya, N., Sergey Krivonogov, Michał Słowiński, Xianyong Cao, Snezhana, Z., 2020. Postglacial history of the Steppe Altai: Climate, fire and plant diversity. *Quat. Sci. Rev.* 249, 106616.
- Zhilich, S., Rudaya, N., Krivonogov, S., Nazarova, L., Pozdnyakov, D., 2017. Environmental dynamics of the Baraba forest-steppe (Siberia) over the last 8000 years and their impact on the types of economic life of the population. *Quat. Sci. Rev.* 163, 152–161.

Of the minor errors, I could not find the reference to Cao et al, 2023, in the Supplementary.

Reply: Thank you very much. Updated.

Fig. S16 shows the biogenic silica profile not only from Lake Tiewaike but also from Lake Kanas.

Reply: Thanks. Revised.

The word pollen is misspelt in the table – polle

Reply: Thanks. Revised.

Reviewer #3 (Remarks to the Author):

Reviewer 3's response to the author's rebuttal.

Thank you very much for taking the time to respond to my remarks. I hope you found them useful. I am happy that you have addressed all of my minor suggestions. Below are my responses to comments 1 -3.

Reply: Many thanks for your positive comments and useful suggestions.

Comment 1 ($\delta^{15}\text{N}$):

Thank you, you have adequately addressed my comment.

Reply: Many thanks for your positive comments.

Comment 2 (Pollen):

I am happy with the pollen diagram, but I would suggest including the meaning A/C in the figure caption. I also suggest that you include some text in the discussion that highlights the significance of *Artemisia* and Chenopodiaceae data and how it fits with your overall story.

Reply: Thank you for your comments. The A/C ratios have been widely used to indicate moisture changes in dryland area with desert or desert-steppe vegetation, but it has some potential problem in its application in mountain forests. The Tiewaike Lake is located in forest vegetation, *Artemisia* and Chenopodiaceae (= Amaranthaceae) pollen from core TWK15A may have originated from local and regional plants with different pollen source area, and their climatic implications are complex. Therefore, we have removed the A/C ratio from this diagram and we instead focus on tree pollen in this sequence.

Comment 3 (Mn/Fe):

Thanks for adding the extra graphics and text. I agree with your discussion and have some minor additions for the text highlighted in red in an attachment for you.

Reply: Many thanks for your edits. Revised.

Reviewer #4 (Remarks to the Author):

The authors have carefully considered the comments, and present new data/figures and discussions to support their conclusions. Overall, I think the data and discussion are more convincing and robust. I have some additional minor comments, which should be addressed easily.

Reply: Many thanks for your positive comments.

1. In Supplementary Information Text, Line 44, please provide the depth of the Tiewaike Lake.

Reply: Thanks. Revised. Please see L47 in Supplementary Information.

2. The Tiewaike Lake is mainly recharged by the precipitation and keeps several meters depth fresh water since ~6.5 ka ago. It is interesting that why a lake with closed lake basin could keep fresh for 6.5 ka; and why its lacustrine sediments are peat like deposits since 8.2 ka ago (Fig. R3). Please clarify these in supplementary or in main text when discuss the Figure S13.

Reply: Thank you for your question. The Tiewaike Lake is an alpine freshwater lake due to its low levels of chemical weathering in the catchment and the relatively low input of soluble chemical substances. Analysis of sediment lithology and organic matter content suggests that the abundance of aquatic plant and organic matter from its upland forest has resulted in the formation of peat sediments over time (Supplementary Fig. 13b). The pollen data of Kanas Lake and peat investigation suggested that the forest vegetation has developed and peat has been accumulating since around 8 kyr (Supplementary Fig. 13b) (e.g., Huang et al., 2018; Cao et al., 2023; Feng et al., 2017). Please see Supplementary Notes 1 and 5 at L43-46 and L193-198.

References:

- Cao, H., Huang X., Xiang L., 2023. Soil erosion caused the increasing Holocene radiocarbon reservoir effect of Lake Kanas in the Altai Mountains. *Radiocarbon* 65, 343–356.
- Feng, Z., Sun, A., Abdusalih, N., Ran, M., Kurban, A., Lan, B., Zhang, D., Yang, Y., 2017. Vegetation changes and associated climatic changes in the southern Altai Mountains within China during the Holocene. *The Holocene* 27, 683–693.

Huang, X., Peng, W., Rudaya, N., Grimm, E.C., Chen, X., Cao, X., Zhang, J., Pan, X., Liu, S., Chen, C., Chen, F., 2018. Holocene Vegetation and Climate Dynamics in the Altai Mountains and Surrounding Areas. *Geophys. Res. Lett.* 45, 6628–6636.